# Sequential Scheduling Method for FJSP with Multi-Objective under Mixed Work Calendars

**Qiang Zeng [1], Menghua Wang [1,\*], Ling Shen [2] and Hongna Song [1]**

[1] School of Energy Science and Engineering, Henan Polytechnic University, Jiaozuo 454000, China; zengqiang@hpu.edu.cn (Q.Z.); songhn@hpu.edu.cn (H.S.)
[2] School of Safety Science and Engineering, Henan Polytechnic University, Jiaozuo 454000, China; shenling@hpu.edu.cn
\* Correspondence: wangmenghua56@163.com; Tel.: +86-183-3755-8911

**Abstract:** A sequential scheduling method for multi-objective, flexible job-shop scheduling problem (FJSP) work calendars is proposed. Firstly, the sequential scheduling problem for the multi-objective FJSP under mixed work calendars was described. Secondly, two key technologies to solve such a problem were proposed: one was a time-reckoning technology based on the machine's work calendar, the other was a sequential scheduling technology. Then, a non-dominated sorting genetic algorithm with an elite strategy (NSGA-II) was designed to solve the problem. In the algorithm, a two-segment encoding method was used to encode the chromosome. A two-segment crossover and mutation operator were used with an improved strategy of genetic operators therein to ensure feasibility of the chromosomes. Time-reckoning technology was used to calculate start and end time of each process. The sequential scheduling technology was used to implement sequential scheduling. The case study shows that the proposed method can obtain an effective Pareto set of the sequential scheduling problem for multi-objective FJSP under mixed work calendars within an acceptable time.

**Keywords:** flexible job-shop scheduling; sequential scheduling; mixed work calendars; multi-objective optimization; NSGA-II

---

## 1. Introduction

In recent years, researchers have shown an increased interest in the field of the flexible job-shop scheduling problem (FJSP), and there have been a large number of related research results. However, there is a phenomenon of mixed work calendars in manufacturing companies, i.e., different machines in the same company or workshop may run under different work calendars. Most of the existing research results cannot be applied when mixed work calendars are considered. Nowadays, because of the increasing pressure in manufacturing companies, more than one objective should be often considered in the process of production scheduling. Just for this reason, multi-objective optimization is more suitable for scheduling requirements. It is often possible to arrange the next batch of scheduling tasks before the completion of the previous one in the production process. This work is called sequential scheduling in this paper. Based on the above reasons, the sequential scheduling for FJSP with multi-objectives under the mixed work calendars has become a new research topic.

There are three main ways to deal with a work calendar in job-shop scheduling: not considering a work calendar, considering a work calendar after scheduling, considering a work calendar in scheduling. Most of the researches belong to the first case [1–3]. Although the calculative complexity is lowest, the scheduling scheme is not associated with the work calendar and is obviously out of touch with practice. A part of the research has been devoted to the second case. The work calendar is not considered in scheduling. After the scheduling scheme is obtained, the time is mapped to the

calendar time through "integral translation" [4]. Although the calculative complexity is low, it is only suitable when the machines operate under the same work calendar. In the third case, the start and stop time of process is arranged according to the work calendar of each machine in scheduling. Although calculative complexity is large, it can ensure that the scheduling scheme is in accordance with practice. Just for this reason, it is suitable for occasions when the machines operate under mixed work calendars. Existing research results on job-shop scheduling with mixed work calendars are rare and there are some shortcomings. Wu Zhijun et al. [5] proposed an implementation strategy and key algorithms of a dynamic work calendar, but did not consider work shift. Huang Yuyue et al. [6] considered work shift in scheduling, but did not consider work system. Stefan et al. [7] considered work system in scheduling, but did not consider work shift. Wan Chunhui et al. [8] not only considered work system but also work shift in scheduling, but there are some shortcomings, such as low operability and large amount of calculation.

In the sequential scheduling process for FJSP with multi-objectives, the scheduling scheme selected by the previous batch of scheduling will affect the machines' initial time status of the next batch of scheduling. In order to maximize utilization of machines and shorten production cycle when arranging the processes, it is necessary to arrange the processes by means of "extrusion insertion". Therefore, how to update the machines' time status according to the selected scheduling scheme, how to construct the machines' initial time status according to the machines' time status and the scheduling start time, and how to arrange the processes by means of "extrusion insertion" based on the machines' work calendars are the three key issues to solve the sequential scheduling for FJSP with multi-objectives under mixed work calendars.

There are two methods to solve FJSP with multi-objectives: indirect method and direct method. The former refers to the method to solve the problem after it is transformed into single objective problem by some means, mainly including the minimax method, linear weighted sum method [9], ideal point method, main object method [10], efficacy coefficient method [11], hierarchical ordering method [12], etc. But this method cannot obtain the entire optimal solution set. The latter refers to the method to obtain the optimal solution set directly in multi-objective space, in which the optimization method based on Pareto optimization thought is most widely used, including the multi-objective genetic algorithm (MOGA) [13], niched pareto genetic algorithm (NPGA) [14], strength Pareto genetic algorithm (SPGA) [15,16], non-dominated sorting genetic algorithm (NSGA) [17], etc. Some research [18,19] has obtained the conclusion that NSGA is superior to other algorithms. With the expansion of the research scope, the shortcomings of NSGA are exposed in the multi-objective optimization algorithms. In order to better solve the multi-objective optimization problem, some scholars proposed a genetic algorithm with non-dominated sorting with elite strategy (NSGA-II). NSGA-II reduces the complexity of NSGA and has the advantages of fast running and good convergence of the solution set. It is one of the most popular multi-objective genetic algorithms at present.

Based on above analysis, a sequential scheduling method for FJSP with multi-objectives under mixed work calendars is proposed in this paper.

## 2. Problem Description

A batch of workpieces needs to be arranged on several machines. Assumptions: ① Each machine runs under the specified work calendar. The work calendar considered in this paper is a combination of work system and work shifts. ② Each workpiece has multiple processes, and there may be more than one feasible machine for each process. ③ The process flow of each workpiece, the adjusting and processing time of each process on the feasible machines, are determined beforehand. ④ When a machine is shut down under its work calendar, the machine stops adjusting, and the process stops processing. The unfinished work will be continued when the machine begins to work again. ⑤ Once a process is started, it should not be interrupted to perform another one. ⑥ When two adjacent batches of workpieces are scheduled, the next schedule should take the machines' time occupation status of the selected scheduling scheme as the initial state. ⑦ The scheduling runs from the specified scheduling

start time. ⑧ The scheduling start time of the next batch is not earlier than that of the previous one. Requirements: to reasonably arrange the processes so as to minimize production cycle and total cost under the above assumptions.

Obviously, the above problem belongs to a sequential scheduling problem for FJSP with multi-objectives under mixed work calendars. Because different machines may operate under different work calendars, the problem is a highly complex NP-hard problem. It can be solved by an intelligent search algorithm. In view of the advantages of a genetic algorithm, taking Excel as the designed platform, this paper designs a non-dominated sorting genetic algorithm with elite strategy (NSGA-II) to solve it.

## 3. Definition of Types, Variables and Arrays

A set of types, variables, and arrays are defined. The types are proc, job, chr, mach, as shown in Figure 1. The variables are shown in Table 1. The arrays are shown in Table 2. chr.R is an array of tpnum × 12. Columns 1 to 12 are used to store task No., workpiece No., process No., machine No., adjustment time, processing time, adjusting start time, adjusting end time, processing start time, processing end time, adjusting cost and processing cost. MA is used to store parameters of each machine. JB is used to store parameters of each workpiece. MMB is used to store the machines' time status before scheduling.

$$proc\begin{cases} \textit{name}(\text{string, process name}) \\ \textbf{\textit{MN}}()(\text{integer, vector of feasible machines' No.}) \\ \textbf{\textit{CT}}()(\text{double, vector of processing time of feasible machines, h}) \\ \textbf{\textit{ST}}()(\text{double, vector of adjuesting time of feasible machines, h}) \\ \textbf{\textit{PC}}()(\text{double, vector of processing cost of feasible machines, yuan/h}) \\ \textbf{\textit{PS}}()(\text{double, vector of adjusting cost of feasible machines, yuan/h}) \end{cases}$$

$$job\begin{cases} \textit{name}(\text{string, workpiece name}) \\ \textit{type}(\text{string, workpiece type}) \\ \textit{pnum}(\text{integer, number of processes}) \\ \textit{dtime}(\text{date, delivery time}) \\ \textit{prc}(\text{double, earlier completion rate, yuan/d}) \\ \textit{dec}(\text{double, delayed completion rate, yuan/d}) \\ \textbf{\textit{PR}}()(\text{proc, process}) \end{cases}$$

$$mach\begin{cases} \textbf{\textit{TS}}()(\textit{variant}, \text{vector of a machine's time status}) \end{cases}$$

$$chr\begin{cases} \textbf{\textit{R}}()(\text{variant, scheduling matrix}) \\ \textbf{\textit{O}}()(\text{double, vector of objectives' value}) \\ \textbf{\textit{MMA}}()(\text{mach, machines' time status after scheduling}) \\ \textit{rank}(\text{integer, frontier value}) \\ \textit{cd}(\text{double, congestion value}) \end{cases}$$

**Figure 1.** Custom type.

**Table 1.** Variables.

| Name | Meaning | Element Type | Category 1 | Category 2 |
|---|---|---|---|---|
| jnum | number of scheduled workpieces | integer | global | input parameter |
| tpnum | number of scheduled processes | integer | global | input parameter |
| mnum | number of scheduled machines | integer | global | input parameter |
| bt | scheduling start time | date | global | input parameter |
| tln | large time | double | global | input parameter |
| popsize | population size | integer | global | input parameter |
| pc | crossover rate | double | global | input parameter |
| pm | mutation rate | double | global | input parameter |
| cr | crossover ratio | double | global | input parameter |
| mr | mutation ratio | double | global | input parameter |
| mbs | number of objectives | integer | global | input parameter |
| maxgen | maximum evolution algebra | integer | global | input parameter |
| nws | number of work systems | integer | global | input parameter |
| nwt | number of work shifts | integer | global | input parameter |
| thr | positive real number close to 0 | double | local | middle parameter |
| epoc | current evolutionary algebra | Integer | local | middle parameter |

**Table 1.** *Cont.*

| Name | Meaning | Element Type | Category 1 | Category 2 |
|---|---|---|---|---|
| *P*1, *P*2 | two parent individuals before crossover operation | chr | local | middle parameter |
| *ch* | individual | chr | local | middle parameter |
| *r* | random real number between 0 and 1 | double | local | middle parameter |
| *st* | adjusting time, h | double | local | middle parameter |
| *ct* | processing time, h | double | local | middle parameter |
| *g* | the earliest adjustable time of a process | date | local | middle parameter |
| *frnum* | number of idle periods of a machine | integer | local | middle parameter |
| *tb* | start time of the *k*th idle period of a machine | date | local | middle parameter |
| *te* | end time of the *k*th idle periods of a machine | date | local | middle parameter |
| *tsb* | adjusting start time of a process | date | local | middle parameter |
| *tse* | adjusting end time of a process | date | local | middle parameter |
| *tcb* | processing start time of a process | date | local | middle parameter |
| *tce* | processing end time of a process | date | local | middle parameter |
| *jc* | details of a process | string | local | middle parameter |

**Table 2.** Arrays.

| Name | Meaning | Type | Size | Type 1 | Type 2 |
|---|---|---|---|---|---|
| **MA** | machine | variant | $mnum \times 5$ | global | input parameter |
| **JB** | workpiece | job | $jnum \times 1$ | global | input parameter |
| **MMB** | machines' time status before scheduling | mach | $mnum \times 1$ | global | input parameter |
| **WS** | work systems | variant | $500 \times 2nws$ | global | input parameter |
| **WT** | work shifts | variant | $8 \times 7nwt$ | global | input parameter |
| **PPOP** | parent population | chr | $Popsize \times 1$ | local | middle parameter |
| **PLPOP** | pairing pool | chr | $popsize/2 \times 1$ | local | middle parameter |
| **OPOP** | population after crossover and mutation operation | chr | $popsize/2 \times 1$ | local | middle parameter |
| **INPOP** | combined population | chr | $popsize \times 1$ | local | middle parameter |
| **CPOP** | offspring population | chr | $popsize \times 1$ | local | middle parameter |
| **OC** | two offspring individuals after crossover operation | chr | $2 \times 1$ | local | middle parameter |

## 4. Key Technologies

### 4.1. Time Reckoning Technology Based on the Machine's Work Calendar

#### 4.1.1. Design and Configuration of Work Calendar

A worksheet "work system" is designed for setting up work systems. In Figure 2, from left to right, every two columns correspond to a work system. The cells of the odd column in the first row are names of work systems. The cells of the odd column in the other rows are days off on the non-weekend. The cells of even columns are work days on the weekend. The dispatcher can also add other work systems to the right as needed.

|     | A         | B         | C         | D         | E         | F   |
|-----|-----------|-----------|-----------|-----------|-----------|-----|
| 1   | X         |           | Y         |           | Z         |     |
| 2   | 2017/1/2  | 2017/1/7  | 2017/1/2  | 2017/1/7  | 2017/1/2  |     |
| 3   | 2017/1/27 | 2017/1/14 | 2017/1/27 | 2017/1/14 | 2017/1/27 |     |
| ... | ...       | ...       | ...       | ...       | ...       | ... |

**Figure 2.** Worksheet "work systems".

A worksheet "work shifts" is designed for setting up work shifts. In Figure 3, columns A to G are setting content of work shift A. Cell A1 is the name of work shift A. Columns A2 to G2 are number of work periods per day from Monday to Sunday. The other rows in each column are used to set up work periods of each day. Taking column A as an example, columns A3 to A8 indicate that the three work periods on Monday are 8:00 to 12:00, 13:00 to 17:00 and 18:00 to 22:00. The dispatcher can also add other work shifts to the right, such as B, C ... as needed.

|   | A     | B     | C     | D     | E     | F     | G     |
|---|-------|-------|-------|-------|-------|-------|-------|
| 1 | A     |       |       |       |       |       |       |
| 2 | 3     | 3     | 3     | 3     | 3     | 1     | 1     |
| 3 | 8:00  | 8:00  | 8:00  | 8:00  | 8:00  | 8:00  | 8:00  |
| 4 | 12:00 | 12:00 | 12:00 | 12:00 | 12:00 | 12:00 | 12:00 |
| 5 | 13:00 | 13:00 | 13:00 | 13:00 | 13:00 |       |       |
| 6 | 17:00 | 17:00 | 17:00 | 17:00 | 17:00 |       |       |
| 7 | 18:00 | 18:00 | 18:00 | 18:00 | 18:00 |       |       |
| 8 | 22:00 | 22:00 | 22:00 | 22:00 | 22:00 |       |       |

**Figure 3.** Worksheet "work shifts".

A worksheet "machines" is designed for configuring the work calendar for each machine. In Figure 4, work system Z and work shift A are assigned to machine "300T", and work system Y and work shift B are assigned to machine "200T".

|     | A           | B            | C            | D           | E          |
|-----|-------------|--------------|--------------|-------------|------------|
| 1   | machine No. | machine code | machine type | work system | work shift |
| 2   | 1           | 300T         | CNC lathe    | Z           | A          |
| 3   | 2           | 200T         | CNC lathe    | Y           | B          |
| ... | ...         | ...          | ...          | ...         | ...        |

**Figure 4.** Worksheet "machines".

4.1.2. Design of Time-Reckoning Function Based on Work Calendar

Based on the design and configuration of the work calendar, six functions were designed to realize time reckoning, namely Isworkday, Nextworkday, Getsd, Forwardwd, Backwd and Getat.

Isworkday: This function has two parameters: *md* (date) and *wds* (string). It is used to judge whether the date *md* is work day or not according to the machine's work system *wds*. If so, return 1, otherwise, return 0. Figure 5 is flow of this function.

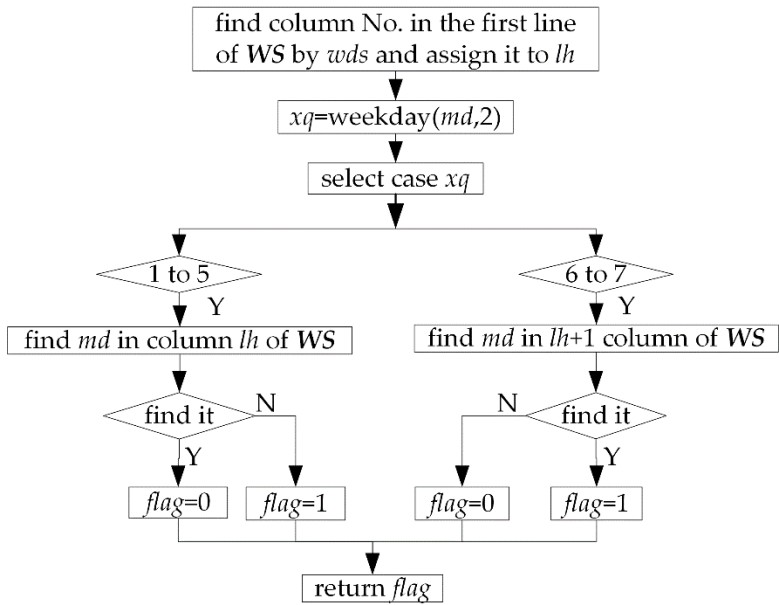

**Figure 5.** Flow of function Isworkday.

Nextworkday: This function has three parameters: *md* (date), *t* (integer) and *wds* (string). It is used to get the work day after *t* days from date *md* according to work system *wds*. *t* > 0 indicates forward reckoning, and *t* < 0 indicates backward reckoning. Figure 6 is the flow of this function.

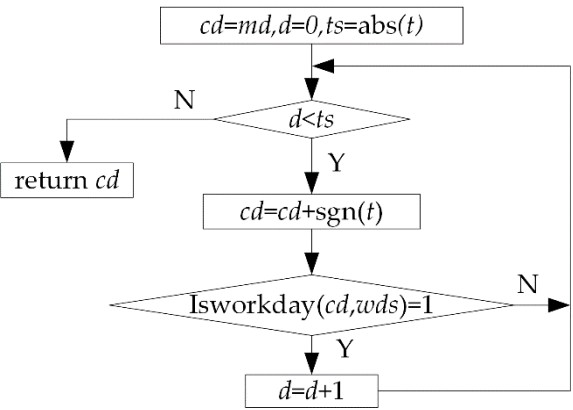

**Figure 6.** Flow of function Nextworkday.

Getsd: This function has three parameters: md (date), t (double), and mn (integer). It is used to get the position of time t on the date md according to work shift of machine mn. Its return value is an array A. A contains 2 elements. A (2) is the flag element. A (2) = 0 indicates that time t belongs to the A (1)th non-work period on the date md of machine mn. A (2) = 1 indicates that time t belongs to the A (1)th work period on the date md of machine mn. In Figure 7, 0:00~24:00 is split into 5 time periods, including 2 work periods 8:00~12:00 (No. 1),13:00~17:00 (No. 2), and 3 non-work periods 0:00~8:00

(No. 0), 12:00~13:00 (No. 1), 17:00~24:00 (No. 2). Figure 8 is flow of this function. The for loop I is used to judge whether time t belongs to the *i*th work period of the date md. If so, let A (1) = i, A (2) = 1, and return A. The for loop II and if statement III are used to judge whether time t belongs to the *i*th non-work period of the date md. If so, let A (1) = i, A (2) = 0, and return A. It is necessary to be pointed that if condition of if statement III is not met, it means that time t does not belong to any of the work periods and non-work periods of 1~wtn, so it can only belong to the 0th non-work period. In this case, the function will return array A, in which A (1) = 0, A (2) = 0.

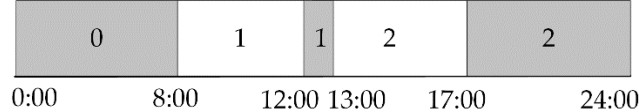

**Figure 7.** Work and non-work periods on one day of a machine.

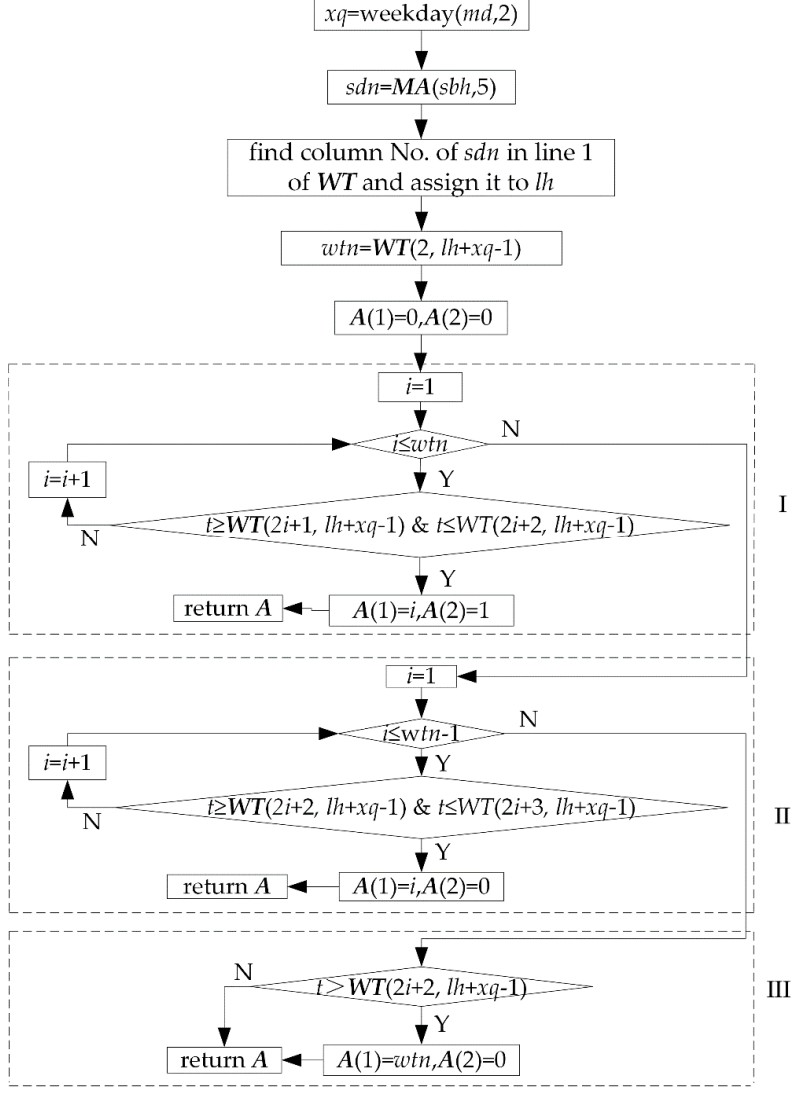

**Figure 8.** Flow of function Getsd.

Forwardwd: This function has three parameters: mdt (date), t (double), and mn (integer). It is used to get the work time after t hours from work time mdt according to work calendar of machine mn by forward reckoning. Figure 9 is a schematic diagram of reckoning process. Figure 10 is flow of this function. In Figure 9, the work period is simply referred to as wp.

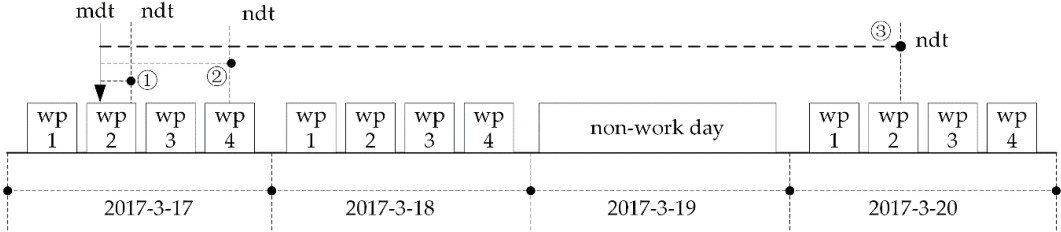

**Figure 9.** Schematic diagram of forward reckoning.

$wds=\textbf{MA}(mn,4)$

$sdn=\textbf{MA}(mn,5)$

find the column No. of *sdn* in line 1 of **WT** and assign it to *lh*

$md=\text{Int}(mdt)$

$mt=24(mdt-md)$

$xq=\text{Weekday}(md,2)$

$wtn=\text{WT}(2,lh+xq-1)$

$A=\text{Getsd}(md,mt,mn)$

$mt + tt\leq \textbf{WT}(2A(1)+2, lh+xq-1)$　　Y　①

$tt=tt-(\textbf{WT}(2A(1)+2, lh+xq-1)-mt)$

$i=A(1)+1$

N　　$i\leq wtn$　③　Y　②　　$i=i+1$

$md=\text{Nextworkday}(md,1,wds)$　　　$\Delta=\textbf{WT}(2i+2, lh+xq-1)-\textbf{WT}(2i+1,lh+xq-1)$　　$tt= tt-\Delta$

$xq= \text{Weekday}(md,2)$

$wtn=\textbf{WT}(2,lh+xq-1)$　　　$\Delta\geq tt$　N

$i=1$　　　Y

N　　$i\leq wtn$　　　$ndt=md+(\textbf{WT}(2i+1, lh+xq-1)+tt)/24$　　$ndt=md+(mt+tt)/24$

Y　　$i=i+1$

$\Delta=\textbf{WT}(2i+2,lh+xq-1)-\textbf{WT}(2i+1,lh+xq-1)$　　$tt=tt-\Delta$

$\Delta\geq tt$　N

Y

$ndt=md+(\textbf{WT}(2i+1,lh+xq-1)+tt)/24$

return *ndt*

**Figure 10.** Flow of function forward.

Backwd: This function has three parameters: mdt (date), t (double) and mn (integer). It is used to get the work time before t hours from work time mdt according to work calendar of machine mn

by backward reckoning. The function Backwd is similar to the function Forwardwd in calculation principle and flow, no more repeat here.

Getat: This function has two parameters: mdt (date) and mn (integer). It is used to get the earliest work time from time mdt according to work calendar of machine mn by forward reckoning. Figure 11 is a schematic diagram of the reckoning process, and Figure 12 is the flow of this function. In Figure 11, the work period is simply referred to as wp.

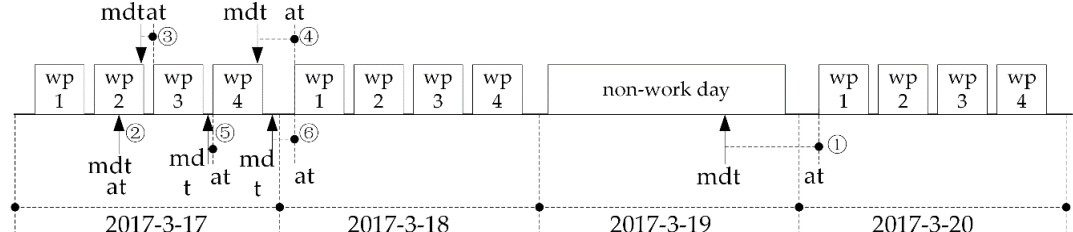

**Figure 11.** Getting the earliest work time.

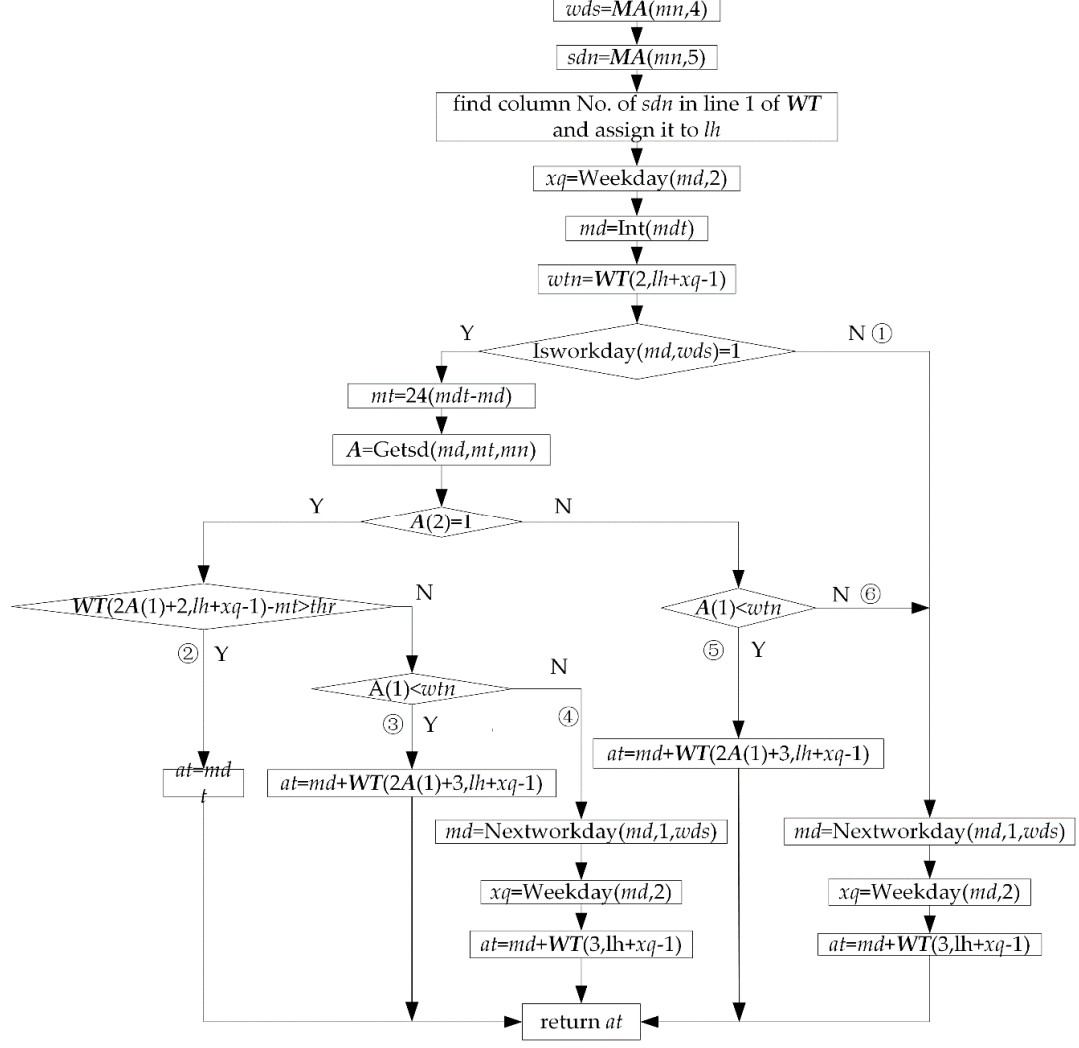

**Figure 12.** Flow of function Getat.

### 4.2. Sequential Scheduling Technology

As mentioned in the introduction, how to construct the machines' initial time status according to the machines' time status and the scheduling start time, and how to arrange the processes by means of

"extrusion insertion" based on the machines' work calendars, are the key issues to solve the sequential scheduling for FJSP with multi-objectives under mixed calendars. The technical steps to solve these key issues in this paper are as follows.

Step 1: Construct machines' initial time status before scheduling and assign them to array *MMB* according to worksheet "machines' time status" and the given scheduling start time.

Construct a worksheet "machines' time status" as shown in Figure 13. Each row is used to store time status information of a machine. Column A is machine No. Columns B and C are start and end time of the first idle period of each machine. Columns C to G are adjusting start time, adjusting end time, details of a process, processing start time, and processing end time of the first process. Columns G and H are start and end time of the second idle period of each machine. Columns H to L are adjusting start time, adjusting end time, details of the second process, processing start time, and processing end time of the second process; and so on.

| | A | B | C | D | E | F | G | H | ... |
|---|---|---|---|---|---|---|---|---|---|
| 1 | 1 | 2017/3/4 8:00 | 2017/3/8 18:23 | 2017/3/8 19:21 | DH:1,J6,2, 2017/3/8 19:21:00-2017/3/9 13:51:00 | 2017/3/8 19:21 | 2017/3/9 13:51 | 2017/3/9 13:51 | ... |
| 2 | 2 | 2017/3/4 8:00 | 2017/3/4 8:00 | 2017/3/4 9:12 | DH:1,J4,4, 2017/3/4 9:12:00-2017/3/6 14:12:00 | 2017/3/4 9:12 | 2017/3/6 14:12 | 2017/3/6 14:12 | ... |
| 3 | .. | ... | ... | ... | ... | ... | ... | ... | ... |

**Figure 13.** Worksheet "machines' time status".

Schematic diagram of constructing the machines' time status before scheduling is shown in Figure 14. Take machine *i* as an example. If the given scheduling start time is in the *k*th idle period of machine *i* (a), let the start time of the first idle period of machine *i* be equal to the given scheduling start time, and the end time of the first idle period of machine *i* be equal to the end time of the *k*th idle period. The subsequent idle periods are taken back in turn. If the given scheduling start time is between the start and end time of a certain process (b), let the start time of the first idle period of machine *i* be equal to the end time of the process, and let the end time of the first idle period of machine *i* be equal to the end time of the first idle period after the process. The subsequent idle periods are taken back in turn.

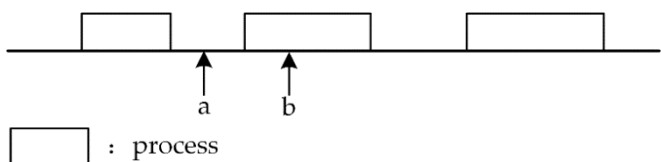

**Figure 14.** Constructing machines' time status before scheduling.

Step 2: Assign *MMB* to the individual's attribute *MMA*.

Step 3: Arrange the processes by means of "extrusion insertion" when the individual is decoded.

The detailed steps are as follows (Figure 15). Firstly, find the first feasible idle period *k* according to time status of machine *i*. Then, insert process *a* into the idle period *k*. Finally, update time status of machine *i*. The method of finding the first feasible idle period is as follows. Determine whether each idle period is enough for process *a* from left to right. If enough, find the first feasible idle period *k*, otherwise continue to search backward. The steps of judging whether the idle period *k* is enough is as follows. Reckon processing end time of process *a* according to the earliest adjustable time, adjusting time, processing time of process *a* and work calendar of the machine. Judge whether the end time of process *a* is not more than the end time of idle period *k*. If so, idle period *k* is a feasible idle period. The steps of updating the time status of machine *i* is as follows. Firstly, increase length of *ch.MMA* (*R* (*i*, 4)).

*TS* by 5. Then, move data from the second datum of *k*th idle period back 5 positions so as to obtain 5 empty positions. Finally, fill adjusting start time, adjusting end time, process details, processing start time, and processing end time of the current process in the 5 empty positions.

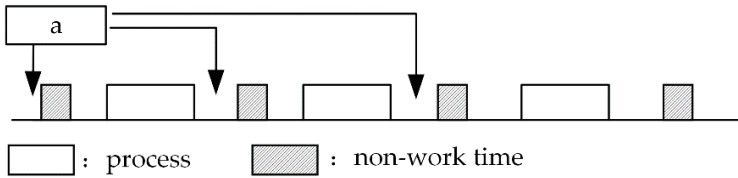

: process      : non-work time

**Figure 15.** Process arrangement by means of extrusion insertion.

Step 4: Output scheduling matrix *R* and the machines' time status array *MMA* of each Pareto solution to worksheet "pareto solution set" at the end of the evolution. Worksheet "pareto solution set" is shown in Figure 16.

| | A | B | C | D | E | F | G | H | I |
|---|---|---|---|---|---|---|---|---|---|
| 1 | 1 | 16 | 211063 | 1 | 1.5(L35MC).1(machining shape) | 5 | 1 | 1 | 300T |
| 2 | 2 | 23 | 205990 | 1 | 1.4(L90GF).1(machining shape) | 4 | 1 | 2 | 200T |
| | **J** | **K** | **F** | **…** | **ADD** | **ADE** | **ADF** | **ADG** | **…** |
| 1 | 0.96 | 9 | 2017/3/6 8:00 | … | 3042 | 32 | 2017/3/4 8:00 | 2017/3/6 8:00 | … |
| 2 | 1.2 | 12 | 2017/3/4 8:00 | … | 2640 | 22 | 2017/3/4 8:00 | 2017/3/6 8:00 | … |
| .. | … | … | … | … | … | … | … | … | … |

**Figure 16.** Worksheet "Pareto solution set".

If a scheduling scheme has two optimization objectives, the columns in worksheet "pareto solution set" are described as follows. Columns A to ADE are data of the Pareto solution. Among them, columns A to D are sequence No., objective 1, objective 2, Pareto frontier value. Columns E to ADD are data of scheduling matrix *R* of the Pareto solution, namely process details, workpiece No., process No., machine No., machine code, adjusting time, processing time, adjusting start time, adjusting end time, processing start time, processing end time, adjusting cost, processing cost. Column ADE is data number of time status of machine 1. The next 32 cells after column ADE are data of time status of machine 1; and so on.

Step 5: Extract scheduling matrix into worksheet "Pareto Solution", and machines' time status into worksheet "machines' time status" from the row double clicked when the dispatcher selects a Pareto solution by double click.

## 5. Non-Dominated Sorting Genetic Algorithm with an Elite Strategy (NSGA-II) Design

### 5.1. Algorithm Flow

Take popsize as an even number, the designed algorithm flow is shown in Figure 17.

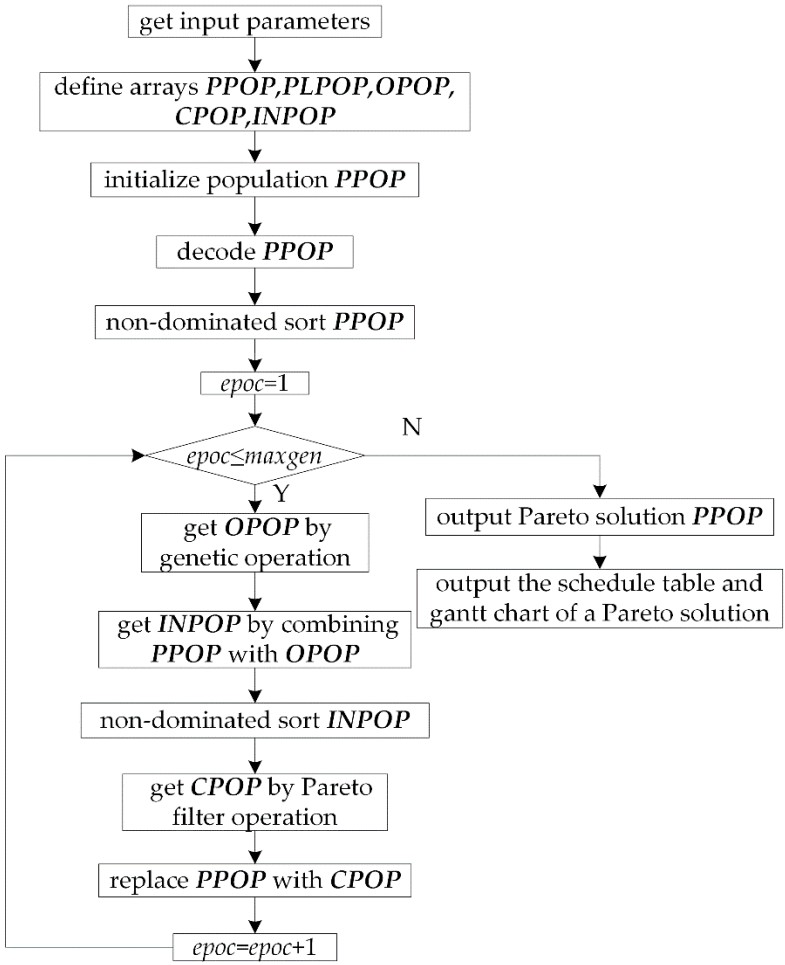

**Figure 17.** Algorithm flowchart.

### 5.2. Obtaining Input Parameters

Firstly, design worksheet "workpiece", "scheduling workpiece", "process flow" and "other parameters" for the dispatcher to set up input parameters. Then, obtain input parameters by the following steps in the algorithm. Read parameters from worksheet "other parameters" and assign them to variables whose type are input parameters in Table 1. Read data from the second row to last row of worksheet "machine" and assign them to array MA. Assign value to array MMB by step (1) in Section 4.2. Assign value to JB by loop: take workpiece i as an example. Firstly, read name, type and number of processes of workpiece i from worksheet "workpiece" and assign them to JB (i).name, JB (i).type and JB (i).pnum. Then, redefine each dimension of JB (i).PR by Redim JB (i).PR (JB (i).pnum). Finally, read parameters of processes of workpiece i from worksheet "process flow" and assign them to JB (i).PR (i)~JB (i).PR (JB (i).pnum) by for loop.

### 5.3. Encoding Method

Encode processes and their processing machines separately by means of "segmentation" as shown in Formula (1). PPOP (i).R is an array of tpnum × 12, in which the second and fourth column are used for encoding, and the other columns are used for auxiliary or decoding. Each gene value of column 2 is a natural number between 1 and jnum. This represents the workpiece No. The number of occurrences

of each natural number is equal to the number of processes of the workpiece. Each gene value of column 4 is the No. of feasible machines for each process.

$$
\boldsymbol{PPOP}(i).\boldsymbol{R} =
\begin{pmatrix}
1 & 2 & 1 & 3 & \cdots & \cdots \\
2 & 1 & 1 & 4 & \cdots & \cdots \\
3 & 1 & 2 & 3 & \cdots & \cdots \\
4 & 2 & 2 & 2 & \cdots & \cdots \\
\cdots & \cdots & \cdots & \cdots & \cdots & \cdots \\
tpnum & 5 & 3 & 4 & \cdots & \cdots
\end{pmatrix}
\tag{1}
$$

### 5.4. Population Initialization

Randomly generate *popsize* individuals and store them in population **PPOP** by flow shown in Figure 18. In Figure 18, the flow to assign feasible machine No. to column 4 of **R** is shown in Figure 19. As can be seen from Figures 18 and 19, the randomly generated individuals are feasible individuals.

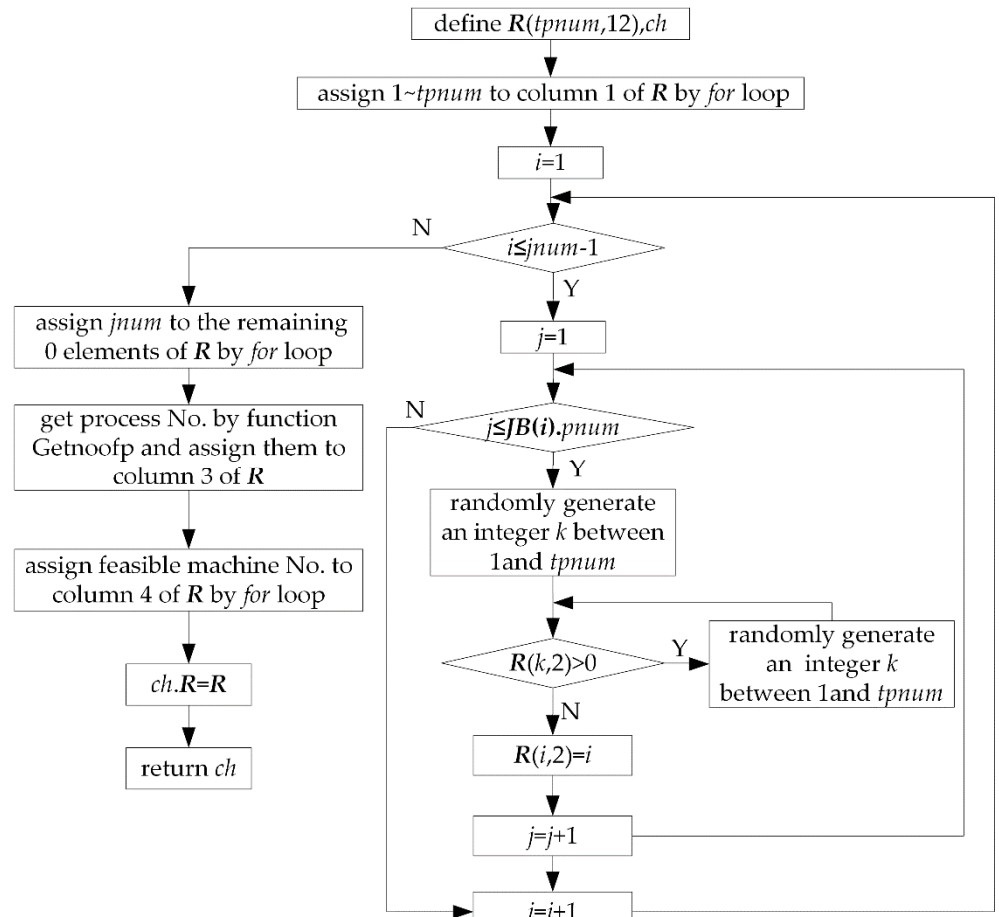

**Figure 18.** Flow of population initialization.

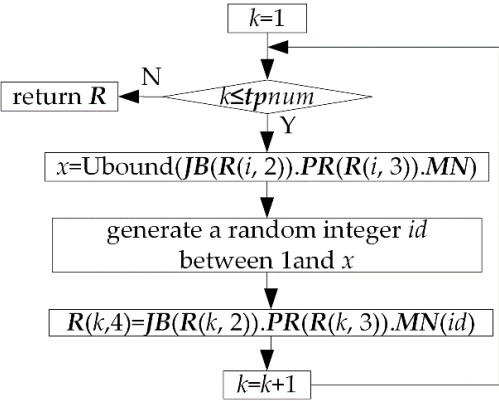

**Figure 19.** Assigning machine No. to column 4 of *R*.

## 5.5. Select Operation

Select "excellent" individuals from parent population **PPOP** by "league mechanism" and form a pairing pool **PLPOP** made up of *popsize*/2 individuals by for loop. Suppose league size is *ts*. Randomly select *ts* individuals from parent population **PPOP** and select an "excellent" individual from them by each individual's frontier value *rank* and congestion degree *cd* into pairing pool **PLPOP** in each loop. Finally, return **PLPOP**.

## 5.6. Genetic Operation

Generate offspring population **OPOP** made up of *popsize* individuals by selection, crossover, and mutation operation from parent population **PPOP** through a genetic operation. The flow is shown in Figure 20.

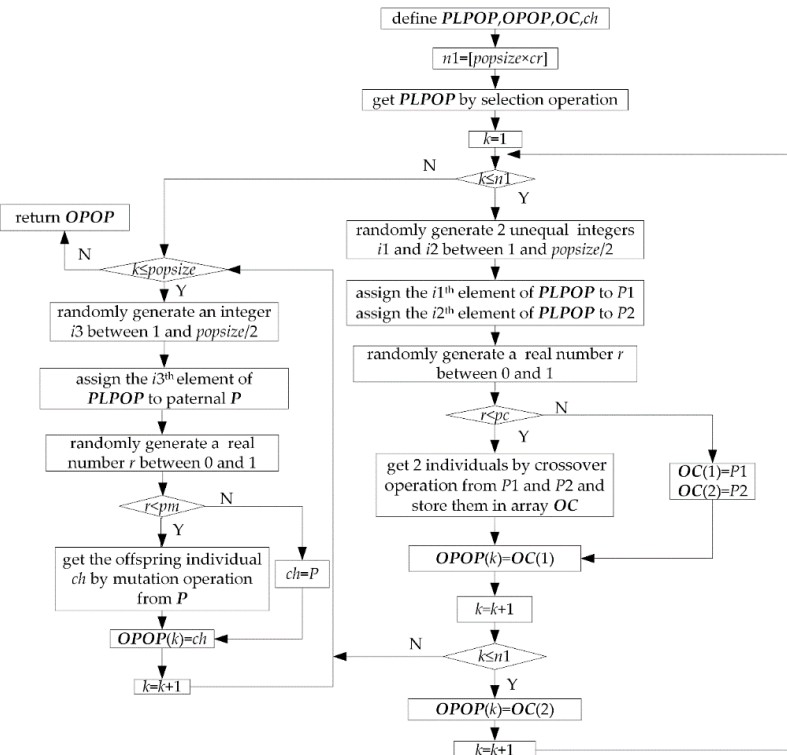

**Figure 20.** Flow of genetic operation.

*5.7. Crossover Operation*

- Crossover operation of processes

Based on an improved strategy of a genetic operator, adopt a kind of crossover method is based on process No. in crossover operation of processes to ensure feasibility of the offspring individuals [20]. Because there is a sequence between processes of the same workpiece, adopt the following measure to ensure this sequence not to be destroyed. Fix the row (workpiece No., process No., and machine No.) of a certain workpiece No. of a parent, and replace the remaining rows of this parent with the other parent's rows except that of this workpiece No. from top to bottom. Take Formula (2) as an example. Fix workpiece 1 in P1.***R***, fix workpiece 3 in P2.***R***, the result of crossover operation between P1.***R*** and P2.***R*** is shown in Formula (3).

$$
P1.\boldsymbol{R} = \begin{pmatrix} 1 & 3 & 1 & 4 \\ 2 & 1 & 1 & 1 \\ 3 & 2 & 1 & 3 \\ 4 & 3 & 2 & 2 \\ 5 & 1 & 2 & 4 \\ 6 & 2 & 2 & 3 \\ 7 & 4 & 1 & 2 \\ 8 & 4 & 2 & 4 \end{pmatrix} \qquad P2.\boldsymbol{R} = \begin{pmatrix} 1 & 2 & 1 & 5 \\ 2 & 3 & 1 & 2 \\ 3 & 1 & 1 & 3 \\ 4 & 1 & 2 & 1 \\ 5 & 4 & 1 & 4 \\ 6 & 2 & 2 & 2 \\ 7 & 4 & 2 & 5 \\ 8 & 3 & 2 & 3 \end{pmatrix} \tag{2}
$$

$$
P1'.\boldsymbol{R} = \begin{pmatrix} 1 & 2 & 1 & 5 \\ 2 & 1 & 1 & 1 \\ 3 & 3 & 1 & 2 \\ 4 & 4 & 1 & 4 \\ 5 & 1 & 2 & 4 \\ 6 & 2 & 2 & 2 \\ 7 & 4 & 2 & 5 \\ 8 & 3 & 2 & 3 \end{pmatrix} \qquad P2'.\boldsymbol{R} = \begin{pmatrix} 1 & 1 & 1 & 1 \\ 2 & 3 & 1 & 2 \\ 3 & 2 & 1 & 3 \\ 4 & 1 & 2 & 4 \\ 5 & 2 & 2 & 3 \\ 6 & 4 & 1 & 2 \\ 7 & 4 & 2 & 4 \\ 8 & 3 & 2 & 3 \end{pmatrix} \tag{3}
$$

- Crossover operation of processing machines

Adopt a kind of two-point crossover method in crossover operation of processing machines. Randomly generate two different integers *mp*1 and *mp*2 between 1 and *tpnum*. If *mp*1 > *mp*2, swamp them to make *mp*1 < *mp*2. Take parent *P1'* as an example. Let *k* change from *mp*1 to *mp*2. For each *k*, find row No. *h* in P2'. ***R*** where the workpiece No. is equal to P1'. ***R*** (*k*, 2) and the process No. is equal to P1'.***R*** (*k*, 3). Let P1'. ***R*** (*k*, 4) = P2'. ***R*** (*h*, 4). Deal with parent P2' by similar method. After crossover operation of processing machines, re-decode P1', P2' and assign them to ***OC*** (1) and ***OC*** (2). For example, if *mp*1 = 3, *mp*2 = 5, the result of crossover operation from *P1'* and *P2'* in Formula (3) is shown in Formula (4).

$$
\boldsymbol{OC}(1).\boldsymbol{R} = \begin{pmatrix} 1 & 2 & 1 & 5 \\ 2 & 1 & 1 & 1 \\ 3 & 3 & 1 & 2 \\ 4 & 4 & 1 & 2 \\ 5 & 1 & 2 & 4 \\ 6 & 2 & 2 & 2 \\ 7 & 4 & 2 & 5 \\ 8 & 3 & 2 & 3 \end{pmatrix} \qquad \boldsymbol{OC}(2).\boldsymbol{R} = \begin{pmatrix} 1 & 1 & 1 & 1 \\ 2 & 3 & 1 & 2 \\ 3 & 2 & 1 & 5 \\ 4 & 1 & 2 & 4 \\ 5 & 2 & 2 & 2 \\ 6 & 4 & 1 & 2 \\ 7 & 4 & 2 & 4 \\ 8 & 3 & 2 & 3 \end{pmatrix} \tag{4}
$$

Obviously, the above crossover operation can ensure feasibility of the offspring individuals.

*5.8. Mutation Operation*

- Mutation operation of processes

Based on an improvement strategy of genetic operator, adopt a kind of sliding mutation method in mutation operation of processes to ensure feasibility of the offspring individuals. Take parent $P$ as an example. Randomly generate an integer $mp$ b9etween 1 and $tpnum$ as the mutation point. Find the nearest index $id$ where $P'R(id,2) = P'R(mp,2)$ from $mp$ upwards and assign it to $s1$. If $id$ is null, let $s1 = 0$. Find the nearest index $id$ where $P'R(id,2) = P'R(mp,2)$ from $mp$ downwards and assign it to $s2$. If $id$ is null, let $s2 = tpnum + 1$. Let $k1 = s1 + 1$. Let $k2 = s2 - 1$. Randomly generate an integer between $k1$ and $k2$, and assign it to $k$. Then slip workpiece No., process No. and machine No. of the $mp^{th}$ row to the $k$th row. Take Formula (5) as an example. If $mp = 3$, $k1 = 2$, $k2 = 8$, $k = 6$, the result of mutation operation is shown in Formula (6).

$$P.R = \begin{pmatrix} 1 & 3 & 1 & 1 \\ 2 & 1 & 1 & 1 \\ 3 & 3 & 2 & 3 \\ 4 & 2 & 1 & 4 \\ 5 & 1 & 2 & 3 \\ 6 & 2 & 2 & 5 \\ 7 & 1 & 3 & 2 \\ 8 & 2 & 3 & 3 \end{pmatrix} \tag{5}$$

$$P'.R = \begin{pmatrix} 1 & 3 & 1 & 1 \\ 2 & 1 & 1 & 1 \\ 3 & 2 & 1 & 4 \\ 4 & 1 & 2 & 4 \\ 5 & 2 & 2 & 5 \\ 6 & 3 & 2 & 3 \\ 7 & 1 & 3 & 2 \\ 8 & 2 & 3 & 3 \end{pmatrix} \tag{6}$$

- Mutation operation of processing machines

Adopt a kind of single-point mutation method in mutation operation of processing machines. Randomly generate an integer $mp$ between 1 and $tpnum$ as the mutation point. Let $k = Ubound$ ($JB(P'.R(mp, 2).PR(P'.R(mp, 3).MN)$). Randomly generate an integer between 1 and $k$ as index of the new processing machine No. and assign it to $id$. Then, let $P'.R(mp,4) = JB(P'.R(mp,2)).PR(P'.R(mp,3)).MN(id)$.

Re-decode individual $P'$ and assign it to $ch$ after mutation operation.

*5.9. Decoding Operation*

The flow of decoding operation is shown in Figure 21.

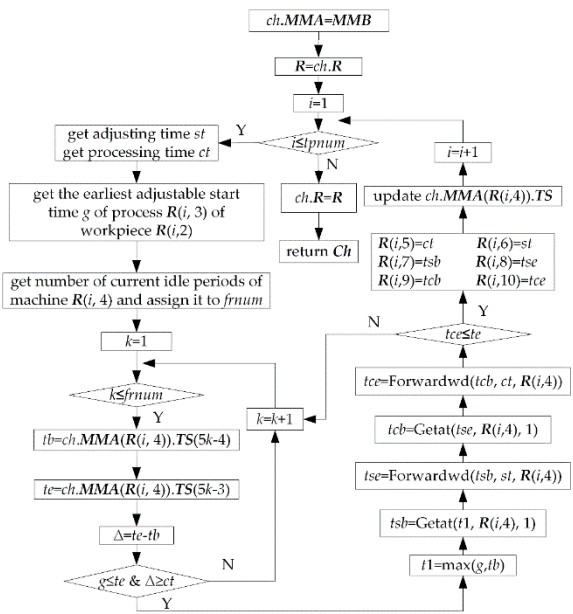

**Figure 21.** Decoding operation.

There are three steps in the decoding operation. Firstly, assign ***MMB*** to *ch.**MMA*** and *ch.**R*** to ***R***. Then, let *i* change from 1 to *tpnum*, arrange processes in sequence and determine columns 5~12 of ***R***. Finally, let *ch.**R*** = ***R***. The following three points need to be explained.

Method of getting machines' adjusting time *st* and processing time *ct* is as follows. Obtain index of machine No. ***R*** (*i*, 4) in ***JB*** (***R*** (*i*, 2). ***PR*** (***R*** (*i*, 3). ***MN*** and assign it to *id*. Let *st* = ***JB*** (***R*** (i, 2). ***PR*** (***R*** (i, 3)). *st* (*id*). Let *ct* = ***JB*** (***R*** (i, 2)). ***PR*** (***R*** (i, 3)). *ct* (id).

The method of getting the earliest adjustable start time *g* of process ***R*** (*i*, 3) of workpiece ***R*** (*i*, 2) is as follows. There are three cases, as shown in Figure 22. Case 1: When ***R*** (*i*, 3) = 1, let *g* = *bt*. Case 2: When ***R*** (*i*, 3) ≠ 1, process ***R*** (*i*, 3) and ***R*** (*i*, 3) − 1 of workpiece ***R*** (*i*, 2) are arranged on the same machine *p*. In this case, process ***R*** (*i*, 3) of workpiece ***R*** (*i*, 2) cannot be adjusted until process ***R*** (*i*, 3) − 1 is completed. So, we can assign the processing end time of process ***R*** (*i*, 3) − 1 of workpiece ***R*** (*i*, 2) to *g*. Assume that row No. of process ***R*** (*i*, 3) − 1 of workpiece ***R*** (*i*, 2) in ***R*** is *h*, let *g* = ***R*** (*h*, 10). Case 3: When ***R*** (*i*, 3) ≠ 1, process ***R*** (*i*, 3) and ***R*** (*i*, 3) − 1 of workpiece ***R*** (*i*, 2) are arranged on different machines. In this case, process ***R*** (*i*, 3) can start its adjusting *st* hours earlier than processing end time of process ***R*** (*i*, 3) − 1 of workpiece ***R*** (*i*, 2). Thus process ***R*** (*i*, 3) can be processed immediately once process ***R*** (*i*, 3) − 1 is finished. Assume that row No. of process ***R*** (*i*, 3) − 1 of workpiece ***R*** (*i*, 2) in ***R*** is *h*, the method to get *g* is as follows. Firstly, get work time *t* according to work calendar of machine ***R*** (*i*, 4) through the forward reckoning function *Getat*, namely let *t* = *Getat* (***R*** (*h*, *10*), R (*i*, 4)). Then, let *g* = *Backwd* (*t*, *st*, ***R*** (*i*,4)).

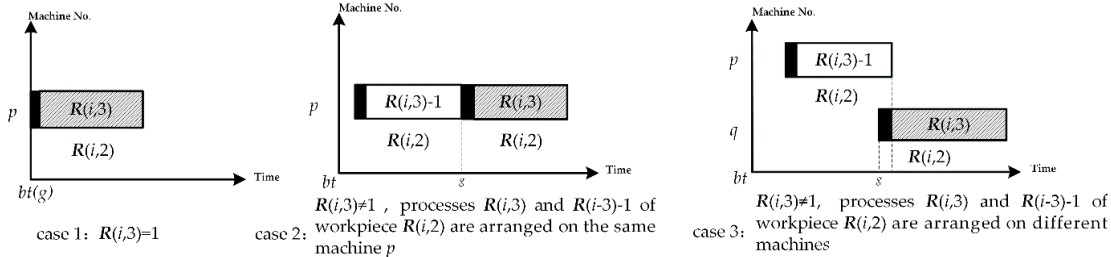

**Figure 22.** Obtaining the earliest adjustable start time *g* for process *R (i, 3)* of workpiece *R (i, 2)*.

The method of updating *ch.**MMA*** (***R*** (*i*, 4)).***TS*** is as follows. In Figure 23, if machine ***R*** (*i*, 4) has been arranged one process *a* in the initial state, initial length of *ch.**MMA*** (***R***(*i*,4)).***TS*** is 7. Namely there

are only 7 elements in *ch*.**MMA** (**R** (*i*, 4)).**TS** which are scheduling start time *bt*, *tsba* (adjusting start time of process *a*), *tsea* (adjusting end time of process *a*), and *jca* (details of process *a*), *tcba* (processing start time of process *a*), tcea (processing end time of process *a*), large time number *tln*. There are two idle periods (Here *frnum* = 2), namely *bt~tsba*, *tcea~tln*. After process *b* is arranged on machine **R** (i, 4) (Assume that process *b* is arranged after processing end time of process *a*), five elements are inserted between *tcea* and *tln*, namely *tsbb* (adjusting start time of process *b*), *tsbe* (adjusting end time of process *b*), *jcb* (details of process *b*), *tcbb* (processing start time of process *b*) and *tceb* (processing end time of process *b*). Length of *ch*.**MMA** (**R** (*i*, 4).**TS** is added to 12. *bt~tln* is split further into 3 idle periods (Here *frnum* = 3), namely *bt~tsba*, *tcea~tsbb*, *tceb~tln*. By above method, with continuous arrangement of each process, the length of *ch*.**MMA** (**R** (*i*, 4)).**TS** is dynamically changed and its elements is dynamically updated. Assume that the insertion period is the *k*th idle period of machine **R** (*i*, 4), then update *ch*.**MMA** (**R** (*i*, 4)).**TS** as follows. Firstly, increase length of *ch*.**MMA** (**R** (*i*, 4)).**TS** by 5. Then, move data from the second datum of the *k*th idle period back 5 positions so as to obtain 5 empty positions. Finally, fill the adjusting start time, adjusting end time, process details, processing start time, and processing end time of the current process in the 5 empty positions.

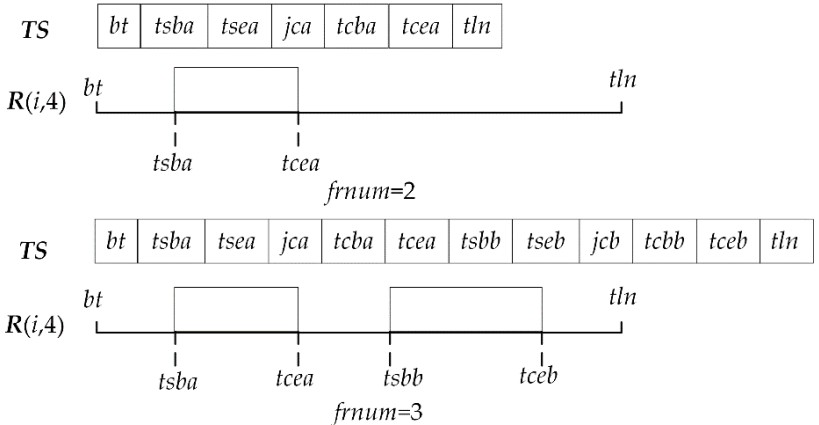

**Figure 23.** Updating *ch.MMA* (*R* (*i*, 4)).*TS*.

*5.10. Calculating Objectives*

- Production cycle (*ch.**O** (1)*)

　　Production cycle is the time span between processing start time and adjusting end time of the scheduled workpieces. Find the minimum value in column 7 of *ch*.**R** and assign it to s1. Find the maximum value in column 10 of *ch*.**R** and assign it to s2. Let *ch*.**O** (1) = s2 − s1.

- Total cost (*ch.**O**(2)*)

　　Total cost is the sum of production cost $c1$, earlier completion cost $c2$, and delayed completion cost $c3$.

　　Production cost ($c1$): Production cost is the sum of adjusting cost and processing cost for all processes of scheduled workpieces. Because column 11 and column 12 of *ch*.**R** are the adjusting cost and processing cost of each process, sum column 11 and column 12, and assign it to $c1$.

　　Earlier completion cost ($c2$): The earlier completion cost of a workpiece is cost produced when the last process of it is finished earlier than its delivery time. It mainly represents storage cost. Sum earlier completion cost of each workpiece to obtain the total earlier completion cost of the scheduled workpieces. Assign it to $c2$. The earlier completion cost of workpiece *i* is equal to the product of earlier completion time and earlier completion fee rate. The method to get earlier completion time of workpiece *i* is as follows: Obtain the difference between delivery time of workpiece *i* and processing

end time of the last process of workpiece *i* in column 10 of *ch*.***R***. Let earlier completion time of workpiece *i* be equal to the maximum value of the difference and 0.

Delayed completion cost (*c*3): The delayed completion cost of a workpiece is cost produced when the last process of it is finished later than its delivery time. It mainly represents penalty fine, crush cost and so on. Sum delayed completion cost of each workpiece to get total delayed completion cost of the scheduled workpieces. Assign it to c3. The delayed completion cost of workpiece *i* is equal to the product of delayed completion time and delayed completion rate. The method to get delayed completion time of workpiece *i* is as follows: Obtain the difference between processing end time of the last process of workpiece *i* in column 10 of ch.***R*** and delivery time of workpiece *i*. Let delayed completion time of workpiece *i* be equal to the maximum value of the difference and 0.

After getting *c*1, *c*2, and *c*3, sum them to *ch*.***O*** (2).

## 6. Case Study

The information of the workpieces to be produced in a job shop is shown in Table 3. The process flow is shown in Table 4. There are 18 machines. The information of the machines is shown in Table 5. There are 3 work systems for the machines which are work system X, Y and Z. Their settings are shown in Figure 24. There are 3 work shifts for the machines which are work shift A, B and C. Their settings are shown in Figure 25.

**Table 3.** The information of the workpieces.

| Workpiece No. | Workpiece Name | Type |
|:---:|:---:|:---:|
| 1 | L2027 | X5 |
| 2 | G46-100F | L5 |
| 3 | ZU30100B2 | M3 |
| 4 | L90GF | M2 |
| 5 | L35MC | X5 |
| 6 | HP6100 | Y2 |
| 7 | 16V32G | M6 |
| 8 | G-45B | N7 |
| 9 | RT-flex | Y1 |
| 10 | DK28 | X8 |

Table 4. Process flow.

| Workpiece Name | Process No. | Process Name | Feasible Machine No. | Processing Time (h) | Adjusting Time (h) | Processing Cost Per Hour (yuan/h) | Adjusting Cost Per Hour (yuan/h) |
|---|---|---|---|---|---|---|---|
| L2027 | 1 | machining shape | 1, 2, 3, 4 | 9, 12, 11.25, 15 | 0.96, 1.2, 1.2, 1.5 | 390, 300, 337, 259 | 336, 240, 297, 212 |
| L2027 | 2 | vehicle end face | 2, 4 | 10, 13 | 1.2, 1.5 | 250, 216 | 200, 177 |
| L2027 | 3 | vehicle circle | 2, 4 | 8, 10 | 1.2, 1.5 | 280, 242 | 240, 212 |
| L2027 | 4 | cone surface | 1, 2, 3, 4 | 5.25, 7, 6.75, 9 | 0.96, 1.2, 1.2, 15 | 390, 300, 337, 259 | 322, 230, 285, 203 |
| L2027 | 5 | drilling and hinge pin holes | 5, 6, 7 | 5.63, 7.5, 10 | 0.64, 0.8, 1 | 507, 390, 300 | 470, 336, 240 |
| L2027 | 6 | milling 20 | 8, 9 | 9, 12 | 0.8, 1 | 390, 300 | 350, 250 |
| L2027 | 7 | grinding middle hole | 10, 11, 12 | 5.63, 7.5, 10 | 0.64, 0.8, 1 | 406, 312, 240 | 353, 252, 180 |
| L2027 | 8 | grinding chamfer | 13, 14 | 6.75, 9 | 0.66, 0.83 | 390, 300 | 336, 240 |
| L2027 | 9 | grinding cylindrical | 15, 16 | 9, 12 | 0.66, 0.83 | 351, 270 | 336, 240 |
| L2027 | 10 | grinding end face | 17, 18 | 7.5, 10 | 0.94, 1.17 | 312, 240 | 252, 180 |
| G46-100F | 1 | machining shape | 1, 2, 3, 4 | 9.75, 13, 10.5, 14 | 0.96, 1.2, 1.2, 1.5 | 390, 300, 314, 241 | 336, 240, 279, 199 |
| G46-100F | 2 | vehicle end face | 2, 4 | 11, 13 | 1.2, 1.5 | 250, 241 | 200, 199 |
| G46-100F | 3 | vehicle circle | 2, 4 | 12, 10 | 1.2, 1.5 | 280, 241 | 240, 199 |
| G46-100F | 4 | cone surface | 1, 2, 3, 4 | 7.5, 10, 8.25, 11 | 1, 1.2, 1.2, 1.5 | 390, 300, 314, 241 | 322, 230, 279, 199 |
| G46-100F | 5 | drilling and hinge pin holes | 5, 6, 7 | 6.75, 9, 12 | 0.64, 0.8, 1 | 507, 390, 300 | 470, 336, 240 |
| G46-100F | 6 | milling 20 | 8, 9 | 9, 12 | 0.8, 1 | 390, 300 | 350, 250 |
| G46-100F | 7 | grinding middle hole | 10, 11, 12 | 4.5, 6, 8 | 0.64, 0.8, 1 | 406, 312, 240 | 353, 252, 180 |
| G46-100F | 8 | grinding chamfer | 13, 14 | 6, 8 | 0.66, 0.83 | 390, 300 | 336, 240 |
| G46-100F | 9 | grinding cylindrical | 15, 16 | 11.25, 15 | 0.66, 0.83 | 351, 270 | 336, 240 |
| G46-100F | 10 | grinding end face | 17, 18 | 6.75, 9 | 0.94, 1.17 | 312, 240 | 252, 180 |
| ZU30100B2 | 1 | machining shape | 2, 4 | 12, 12 | 1.2, 1.5 | 300, 241 | 240, 199 |
| ZU30100B2 | 2 | vehicle end fac | 1, 2, 4 | 7.5, 10, 10 | 0.96, 1.2, 1.5 | 325, 250, 241 | 280, 200, 199 |
| ZU30100B2 | 3 | vehicle circle | 1, 2, 4 | 6, 8, 13 | 0.96, 1.2, 1.5 | 364, 280, 241 | 336, 240, 199 |
| ZU30100B2 | 4 | cone surface | 2, 4 | 7, 15 | 1.2, 1.5 | 300, 241 | 230, 199 |
| ZU30100B2 | 5 | drilling and hinge pin holes | 5, 6, 7 | 6.75, 9, 12 | 0.64, 0.8, 1 | 507, 390, 300 | 470, 336, 240 |
| ZU30100B2 | 6 | milling 20 | 8, 9 | 12, 16 | 0.8, 1 | 390, 300 | 350, 250 |
| ZU30100B2 | 7 | grinding middle hole | 10, 11, 12 | 6.75, 9, 12 | 0.64, 0.8, 1 | 406, 312, 240 | 353, 252, 180 |
| ZU30100B2 | 8 | grinding chamfer | 13, 14 | 7.5, 10 | 0.66, 0.83 | 390, 300 | 336, 240 |
| ZU30100B2 | 9 | grinding cylindrical | 15, 16 | 9.75, 13 | 0.66, 0.83 | 351, 270 | 336, 240 |
| ZU30100B2 | 10 | grinding end face | 17, 18 | 8.25, 11 | 0.94, 1.17 | 312, 240 | 252, 180 |

**Table 4.** *Cont.*

| Workpiece Name | Process No. | Process Name | Feasible Machine No. | Processing Time (h) | Adjusting Time (h) | Processing Cost Per Hour (yuan/h) | Adjusting Cost Per Hour (yuan/h) |
|---|---|---|---|---|---|---|---|
| L90GF | 1 | machining shape | 1, 2, 3, 4 | 9, 12, 11.25, 15 | 0.96, 1.2, 1.2, 1.5 | 390, 300, 314, 241 | 336, 240, 279, 199 |
| L90GF | 2 | vehicle end face | 1, 2, 3, 4 | 7.5, 10, 9.75, 13 | 0.96, 1.2, 1.2, 1.5 | 325, 250, 314, 241 | 280, 200, 279, 199 |
| L90GF | 3 | vehicle circle | 3, 4 | 9, 12 | 1.2, 1.5 | 314, 241 | 279, 199 |
| L90GF | 4 | cone surface | 1, 2, 3, 4 | 9, 12, 9.75, 13 | 0.96, 1.2, 1.2, 1.5 | 390, 300, 314, 241 | 322, 230, 279, 199 |
| L90GF | 5 | drilling and hinge pin holes | 5, 6, 7 | 8.44, 11.25, 15 | 0.64, 0.8, 1 | 507, 390, 300 | 470, 336, 240 |
| L90GF | 6 | milling 20 | 8, 9 | 9, 12 | 0.8, 1 | 390, 300 | 350, 250 |
| L90GF | 7 | grinding middle hole | 10, 11, 12 | 5.06, 6.75, 9 | 0.64, 0.8, 1 | 406, 312, 240 | 353, 252, 180 |
| L90GF | 8 | grinding chamfer | 13, 14 | 8.25, 11 | 0.66, 0.83 | 390, 300 | 336, 240 |
| L90GF | 9 | grinding cylindrical | 15, 16 | 10.5, 14 | 0.66, 0.83 | 351, 270 | 336, 240 |
| L90GF | 10 | grinding end face | 17, 18 | 9, 12 | 0.94, 1.17 | 312, 240 | 252, 180 |
| L35MC | 1 | machining shape | 1, 2, 3, 4 | 9, 12, 11.25, 15 | 0.96, 1.2, 1.2, 1.5 | 390, 300, 314, 241 | 336, 240, 279, 199 |
| L35MC | 2 | vehicle end face | 3, 4 | 9.75, 13 | 1.2, 1.5 | 314, 241 | 279, 199 |
| L35MC | 3 | vehicle circle | 3, 4 | 7.5, 10 | 1.2, 1.5 | 314, 241 | 279, 199 |
| L35MC | 4 | cone surface | 1, 2, 3, 4 | 9, 12, 10.5, 14 | 0.96, 1.2, 1.2, 1.5 | 390, 300, 314, 241 | 322, 230, 279, 199 |
| L35MC | 5 | drilling and hinge pin holes | 5, 6, 7 | 6.19, 8.25, 11 | 0.64, 0.8, 1 | 507, 390, 300 | 470, 336, 240 |
| L35MC | 6 | milling 20 | 8, 9 | 6.75, 9 | 0.8, 1 | 390, 300 | 350, 250 |
| L35MC | 7 | grinding middle hole | 10, 11, 12 | 4.5, 6, 8 | 0.64, 0.8, 1 | 406, 312, 240 | 353, 252, 180 |
| L35MC | 8 | grinding chamfer | 13, 14 | 9, 12 | 0.66, 0.83 | 390, 300 | 336, 240 |
| L35MC | 9 | grinding cylindrical | 15, 16 | 9, 12 | 0.66, 0.83 | 351, 270 | 336, 240 |
| L35MC | 10 | grinding end face | 17, 18 | 9.75, 13 | 0.94, 1.17 | 312, 240 | 252, 180 |
| HP6100 | 1 | machining shape | 1, 2, 3, 4 | 9, 12, 11.25, 15 | 0.96, 12, 1.2, 1.5 | 390, 300, 314, 241 | 336, 240, 279, 199 |
| HP6100 | 2 | vehicle end face | 1, 2 | 7.5, 10 | 0.96, 1.2 | 325, 250 | 280, 200 |
| HP6100 | 3 | vehicle circle | 1, 2 | 6, 8 | 0.96, 1.2 | 364, 280 | 336, 240 |
| HP6100 | 4 | cone surface | 1, 2, 3, 4 | 11.25, 15, 11.25, 15 | 0.96, 1.2, 1.2, 1.5 | 390, 300, 314, 241 | 322, 230, 279, 199 |
| HP6100 | 5 | drilling and hinge pin holes | 5, 6, 7 | 6.75, 9, 12 | 0.64, 0.8, 1 | 507, 390, 300 | 470, 336, 240 |
| HP6100 | 6 | milling 20 | 8, 9 | 9, 12 | 0.8, 1 | 390, 300 | 350, 250 |
| HP6100 | 7 | grinding middle hole | 10, 11, 12 | 4.78, 6.38, 8.5 | 0.64, 0.8, 1 | 406, 312, 240 | 353, 252, 180 |
| HP6100 | 8 | grinding chamfer | 13, 14 | 7.5, 10 | 0.66, 0.83 | 390, 300 | 336, 240 |
| HP6100 | 9 | grinding cylindrical | 15, 16 | 8.25, 11 | 0.66, 0.83 | 351, 270 | 336, 240 |
| HP6100 | 10 | grinding end face | 17, 18 | 8.25, 11 | 0.94, 1.17 | 312, 240 | 252, 180 |

**Table 4.** *Cont.*

| Workpiece Name | Process No. | Process Name | Feasible Machine No. | Processing Time (h) | Adjusting Time (h) | Processing Cost Per Hour (yuan/h) | Adjusting Cost Per Hour (yuan/h) |
|---|---|---|---|---|---|---|---|
| 16V32G | 1 | machining shape | 1, 2 | 9, 12 | 0.96, 1.2 | 390, 300 | 336, 240 |
| 16V32G | 2 | vehicle end face | 1, 2 | 7.5, 10 | 0.96, 1.2 | 325, 250 | 280, 200 |
| 16V32G | 3 | vehicle circle | 2, 3, 4 | 12, 7.5, 10 | 1.2, 1.2, 1.5 | 280, 314, 241 | 240, 279, 199 |
| 16V32G | 4 | cone surface | 2, 3, 4 | 12, 11.25, 15 | 1.2, 1.2, 1.5 | 300, 314, 241 | 230, 279, 199 |
| 16V32G | 5 | drilling and hinge pin holes | 5, 6, 7 | 8.44, 11.25, 15 | 0.64, 0.8,1 | 507, 390, 300 | 470, 336, 240 |
| 16V32G | 6 | milling 20 | 8, 9 | 7.5, 10 | 0.8, 1 | 390, 300 | 350, 250 |
| 16V32G | 7 | grinding middle hole | 10, 11, 12 | 6.08, 8.1, 9 | 0.74, 0.92, 1 | 374, 288, 240 | 290, 207, 180 |
| 16V32G | 8 | grinding chamfer | 13, 14 | 9, 12 | 0.66, 0.83 | 390, 300 | 336, 240 |
| 16V32G | 9 | grinding cylindrical | 15, 16 | 10.5, 14 | 0.66, 0.83 | 351, 270 | 336, 240 |
| 16V32G | 10 | grinding end face | 17, 18 | 8.25, 11 | 0.94, 1.17 | 312, 240 | 252, 180 |
| G-45B | 1 | machining shape | 2, 4 | 8, 10 | 1.2, 1.5 | 300, 241 | 240, 199 |
| G-45B | 2 | vehicle end face | 1, 2, 3, 4 | 7.5, 10, 9.75, 13 | 0.96, 1.2, 1.2, 1.5 | 325, 250, 314, 241 | 280, 200, 279, 199 |
| G-45B | 3 | vehicle circle | 2, 4 | 8, 10 | 1.2, 1.5 | 280, 241 | 240, 199 |
| G-45B | 4 | cone surface | 1, 2, 3, 4 | 8.25, 11, 11.25, 15 | 0.96, 1.2, 1.2, 1.5 | 390, 300, 314, 241 | 322, 230, 279, 199 |
| G-45B | 5 | drilling and hinge pin holes | 5, 6, 7 | 7.31, 9.75, 13 | 0.64, 0.8, 1 | 338, 260, 200 | 314, 224, 160 |
| G-45B | 6 | milling 20 | 8, 9 | 8.25, 11 | 0.8, 1 | 390, 300 | 350, 250 |
| G-45B | 7 | grinding middle hole | 10, 11, 12 | 7.43, 9.9, 11 | 0.74, 0.92, 1 | 374, 288, 240 | 290, 207, 180 |
| G-45B | 8 | grinding chamfer | 13, 14 | 9, 12 | 0.66, 0.83 | 390, 300 | 336, 240 |
| G-45B | 9 | grinding cylindrical | 15, 16 | 7.5, 10 | 0.66, 0.83 | 351, 270 | 336, 240 |
| G-45B | 10 | grinding end face | 17, 18 | 7.5, 10 | 0.94, 1.17 | 312, 240 | 252, 180 |

**Table 5.** The information of the machines.

| Machine No. | Machine Code | Machine Type | Work System | Work Shift |
|---|---|---|---|---|
| 1 | 300T | NC lathe | X | A |
| 2 | 200T | NC lathe | Y | B |
| 3 | T52 | NC lathe | Z | A |
| 4 | T42 | NC lathe | Y | C |
| 5 | Z6018 | bench drill | X | A |
| 6 | Z5018 | bench drill | X | A |
| 7 | Z4018 | bench drill | Y | B |
| 8 | X8126 | milling machine | Z | C |
| 9 | X5126 | milling machine | Y | B |
| 10 | M5515 | horizontal grinding machine tool | X | A |
| 11 | M4515 | horizontal grinding machine tool | X | C |
| 12 | M3515 | horizontal grinding machine tool | Y | B |
| 13 | J5001 | semi-automatic cylindrical grinder | Z | A |
| 14 | J4001 | semi-automatic cylindrical grinder | Y | B |
| 15 | 3U5 | cylindrical grinder | X | C |
| 16 | 2U5 | cylindrical grinder | X | A |
| 17 | 120CNC | internal grinder | Y | B |
| 18 | 111CNC | internal grinder | Z | C |

|  | A | B | C | D | E | F |
|---|---|---|---|---|---|---|
| 1 | **X** |  | **Y** |  | **Z** |  |
| 2 | 2017/1/2 | 2017/1/7 | 2017/1/2 | 2017/1/7 | 2017/1/2 |  |
| 3 | 2017/1/27 | 2017/1/14 | 2017/1/27 | 2017/1/14 | 2017/1/27 |  |
| 4 | 2017/1/30 | 2017/1/21 | 2017/1/30 | 2017/1/21 | 2017/1/30 |  |
| 5 | 2017/1/31 | 2017/2/4 | 2017/1/31 | 2017/2/4 | 2017/1/31 |  |
| 6 | 2017/2/1 | 2017/2/18 | 2017/2/1 | 2017/2/18 | 2017/2/1 |  |
| 7 | 2017/2/2 | 2017/2/25 | 2017/2/2 | 2017/2/25 | 2017/2/2 |  |
| 8 | 2017/4/3 | 2017/3/4 | 2017/4/3 | 2017/3/4 | 2017/4/3 |  |
| 9 | 2017/4/4 | 2017/3/11 | 2017/4/4 | 2017/3/11 | 2017/4/4 |  |
| 10 | 2017/5/1 | 2017/3/18 | 2017/5/1 | 2017/3/18 | 2017/5/1 |  |
| 11 | 2017/5/29 | 2017/3/25 | 2017/5/29 | 2017/3/25 | 2017/5/29 |  |
| 12 | 2017/5/30 | 2017/4/1 | 2017/5/30 | 2017/4/1 | 2017/5/30 |  |
| 13 | 2017/10/2 | 2017/4/8 | 2017/10/2 | 2017/4/8 | 2017/10/2 |  |
| 14 | 2017/10/3 | 2017/4/15 | 2017/10/3 | 2017/4/15 | 2017/10/3 |  |
| 15 | 2017/10/4 | 2017/4/22 | 2017/10/4 | 2017/4/22 | 2017/10/4 |  |
| 16 | 2017/10/5 | 2017/5/6 | 2017/10/5 | 2017/5/6 | 2017/10/5 |  |
| 17 | 2017/10/6 | 2017/5/13 | 2017/10/6 | 2017/5/13 | 2017/10/6 |  |
| 18 |  | 2017/5/20 |  | 2017/5/20 |  |  |
| ... |  | ... |  | ... |  |  |
| 49 |  | ... |  | 2017/12/30 |  |  |
| ... |  | ... |  |  |  |  |
| 95 |  | 2017/12/31 |  |  |  |  |

**Figure 24.** Settings of worksheet "work system".

|  | A | B | C | D | E | F | G | H | I | J | K |
|---|---|---|---|---|---|---|---|---|---|---|---|
| 1 | A |  |  |  |  |  |  | B |  |  |  |
| 2 | 3 | 3 | 3 | 3 | 3 | 1 | 1 | 2 | 2 | 2 | 2 |
| 3 | 8:00 | 8:00 | 8:00 | 8:00 | 8:00 | 8:00 | 8:00 | 8:00 | 8:00 | 8:00 | 8:00 |
| 4 | 12:00 | 12:00 | 12:00 | 12:00 | 12:00 | 12:00 | 12:00 | 12:00 | 12:00 | 12:00 | 12:00 |
| 5 | 13:00 | 13:00 | 13:00 | 13:00 | 13:00 |  |  | 13:00 | 13:00 | 13:00 | 13:00 |
| 6 | 17:00 | 17:00 | 17:00 | 17:00 | 17:00 |  |  | 17:00 | 17:00 | 17:00 | 17:00 |
| 7 | 18:00 | 18:00 | 18:00 | 18:00 | 18:00 |  |  |  |  |  |  |
| 8 | 22:00 | 22:00 | 22:00 | 22:00 | 22:00 |  |  |  |  |  |  |

|  | L | M | N | O | P | Q | R | S | T | U |  |
|---|---|---|---|---|---|---|---|---|---|---|---|
| 1 |  |  |  | C |  |  |  |  |  |  |  |
| 2 | 2 | 2 | 2 | 3 | 3 | 3 | 3 | 3 | 3 | 3 |  |
| 3 | 8:00 | 8:00 | 8:00 | 0:00 | 0:00 | 0:00 | 0:00 | 0:00 | 0:00 | 0:00 |  |
| 4 | 12:00 | 12:00 | 12:00 | 8:00 | 8:00 | 8:00 | 8:00 | 8:00 | 8:00 | 8:00 |  |
| 5 | 13:00 | 13:00 | 13:00 | 9:00 | 9:00 | 9:00 | 9:00 | 9:00 | 9:00 | 9:00 |  |
| 6 | 17:00 | 17:00 | 17:00 | 12:00 | 12:00 | 12:00 | 12:00 | 12:00 | 12:00 | 12:00 |  |
| 7 |  |  |  | 13:00 | 13:00 | 13:00 | 13:00 | 13:00 | 13:00 | 13:00 |  |
| 8 |  |  |  | 17:00 | 17:00 | 17:00 | 17:00 | 17:00 | 17:00 | 17:00 |  |

**Figure 25.** Settings of worksheet "work shifts".

- No. 1 schedule

Start time of No. 1 schedule is 2017/3/4 8:00:00. It aims to arrange processing tasks of three workpieces on 18 machines. The information of the scheduled workpieces is shown in Table 6. Initial time status of the machines is shown in Figure 26. Other parameters are shown in Table 7.

**Table 6.** The information of the scheduled workpieces of No. 1 schedule.

| Workpiece No. | Workpiece Name | Type | Total Process Number | Delivery Time | Earlier Completion Rate (yuan /d) | Delayed Completion Rate (yuan/d) |
|---|---|---|---|---|---|---|
| 1 | L2027 | X5 | 10 | 2017/4/16 | 100 | 1000 |
| 2 | G46-100F | L5 | 10 | 2017/4/15 | 150 | 1200 |
| 3 | ZU30100B2 | M3 | 10 | 2017/5/13 | 80 | 800 |

|  | A | B | C |
|---|---|---|---|
| 1 | 1 | 2017/3/4 8:00 | 4637/11/26 0:00 |
| 2 | 2 | 2017/3/4 8:00 | 4637/11/26 0:00 |
| … | … | … | … |
| 17 | 17 | 2017/3/4 8:00 | 4637/11/26 0:00 |
| 18 | 18 | 2017/3/4 8:00 | 4637/11/26 0:00 |

**Figure 26.** Setting of worksheet "initial time status of the machines".

**Table 7.** Other parameters.

| Parameter | Value | Parameter | Value |
|---|---|---|---|
| *jnum* | 7 | *cr* | 0.7 |
| *tpnum* | 42 | *mr* | 0.3 |
| *mnum* | 10 | *maxgen* | 200 |
| *bt* | 2017/3/4 8:00:00 | *mbs* | 2 |
| *tln* | 4637/11/26 0:00:00 | *nws* | 3 |
| *popsize* | 40 | *nwt* | 3 |
| *pc* | 0.5 | *thr* | 0.001 |
| *pm* | 0.1 | | |

Let the algorithm run independently 20 times. The calculation time of each evolutionary calculation is about 40 s. Each evolutionary calculation can obtain almost the same and uniform Pareto solution set. This shows that convergence effect is good. The Pareto solution set obtained by one of the evolutionary calculations is shown in Figure 27. The dispatcher can select one of the Pareto solutions to arrange production. The schedule table of a Pareto solution (production cycle is 12.28 days, total cost is 105,226.84 yuan) is shown in Table 8. A Gantt chart of the scheduled workpieces and a Gantt chart of the machines are shown in Figures 28 and 29.

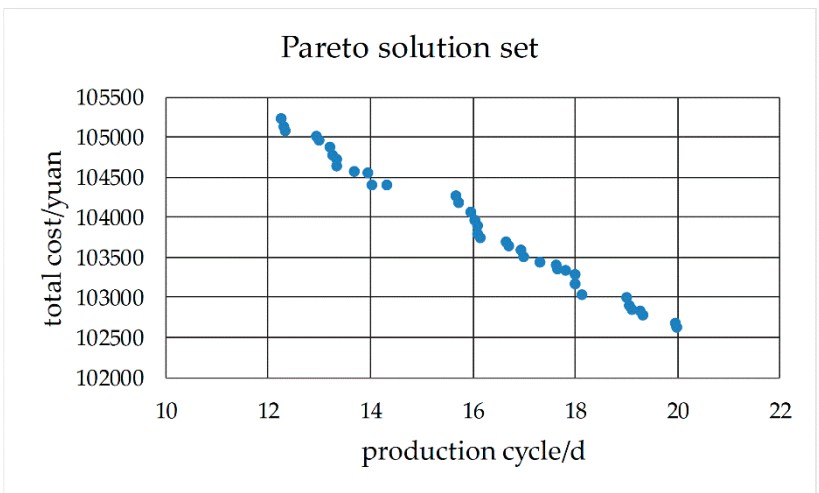

**Figure 27.** Pareto solution set of No. 1 schedule.

**Table 8.** Schedule table of a Pareto solution.

| No. | Workpiece No. | Process No. | Machine No. | Adjusting Time (h) | Processing Time (h) | Adjusting Start Time | Adjusting End Time | Processing Start Time | Processing End Time | Adjusting Cost (yuan) | Processing Cost (yuan) |
|---|---|---|---|---|---|---|---|---|---|---|---|
| 1 | 3 | 1 | 4 | 1.5 | 12 | 2017/3/4 9:00 | 2017/3/4 10:30 | 2017/3/4 10:30 | 2017/3/6 6:30 | 299.01 | 2895.69 |
| 2 | 3 | 2 | 4 | 1.5 | 10 | 2017/3/6 6:30 | 2017/3/6 8:00 | 2017/3/6 9:00 | 2017/3/7 3:00 | 299.01 | 2413.07 |
| 3 | 1 | 1 | 1 | 0.96 | 9 | 2017/3/6 8:00 | 2017/3/6 8:57 | 2017/3/6 8:57 | 2017/3/6 19:57 | 322.56 | 3510.00 |
| 4 | 1 | 2 | 2 | 1.2 | 10 | 2017/3/6 15:48 | 2017/3/6 17:00 | 2017/3/7 8:00 | 2017/3/8 10:00 | 240.00 | 2500.00 |
| 5 | 3 | 3 | 1 | 0.96 | 6 | 2017/3/6 21:02 | 2017/3/6 22:00 | 2017/3/7 8:00 | 2017/3/7 15:00 | 322.56 | 2184.00 |
| 6 | 3 | 4 | 2 | 1.2 | 7 | 2017/3/8 10:00 | 2017/3/8 11:12 | 2017/3/8 11:12 | 2017/3/9 10:12 | 276.00 | 2100.00 |
| 7 | 2 | 1 | 4 | 1.5 | 14 | 2017/3/7 3:00 | 2017/3/7 4:30 | 2017/3/7 4:30 | 2017/3/8 3:30 | 299.01 | 3378.30 |
| 8 | 2 | 2 | 2 | 1.2 | 11 | 2017/3/9 10:12 | 2017/3/9 11:24 | 2017/3/9 11:24 | 2017/3/10 15:24 | 240.00 | 2750.00 |
| 9 | 3 | 5 | 5 | 0.64 | 6.75 | 2017/3/9 9:33 | 2017/3/9 10:12 | 2017/3/9 10:12 | 2017/3/9 18:57 | 301.06 | 3422.25 |
| 10 | 2 | 3 | 4 | 1.5 | 10 | 2017/3/10 13:54 | 2017/3/10 15:24 | 2017/3/10 15:24 | 2017/3/11 9:24 | 299.01 | 2413.07 |
| 11 | 3 | 6 | 8 | 0.8 | 12 | 2017/3/9 16:12 | 2017/3/9 17:00 | 2017/3/10 0:00 | 2017/3/10 14:00 | 280.00 | 4680.00 |
| 12 | 1 | 3 | 4 | 1.5 | 10 | 2017/3/8 11:30 | 2017/3/8 14:00 | 2017/3/8 14:00 | 2017/3/9 7:00 | 318.10 | 2421.72 |
| 13 | 2 | 4 | 4 | 1.5 | 11 | 2017/3/11 9:24 | 2017/3/11 10:54 | 2017/3/11 10:54 | 2017/3/13 5:54 | 299.01 | 2654.38 |
| 14 | 3 | 7 | 10 | 0.64 | 6.75 | 2017/3/10 13:21 | 2017/3/10 14:00 | 2017/3/10 14:00 | 2017/3/10 21:45 | 225.79 | 2737.80 |
| 15 | 2 | 5 | 5 | 0.64 | 6.75 | 2017/3/10 21:21 | 2017/3/10 22:00 | 2017/3/13 8:00 | 2017/3/13 15:45 | 301.06 | 3422.25 |
| 16 | 1 | 4 | 1 | 0.96 | 5.25 | 2017/3/8 21:02 | 2017/3/8 22:00 | 2017/3/9 8:00 | 2017/3/9 14:15 | 309.12 | 2047.50 |
| 17 | 1 | 5 | 5 | 0.64 | 5.625 | 2017/3/9 18:57 | 2017/3/9 19:35 | 2017/3/9 19:35 | 2017/3/10 11:12 | 301.06 | 2851.88 |
| 18 | 1 | 6 | 8 | 0.8 | 9 | 2017/3/10 14:00 | 2017/3/10 14:48 | 2017/3/10 14:48 | 2017/3/11 6:48 | 280.00 | 3510.00 |
| 19 | 1 | 7 | 10 | 0.64 | 5.625 | 2017/3/10 21:45 | 2017/3/13 8:23 | 2017/3/13 8:23 | 2017/3/13 15:00 | 225.79 | 2281.50 |
| 20 | 1 | 8 | 13 | 0.664 | 6.75 | 2017/3/13 14:21 | 2017/3/13 15:00 | 2017/3/13 15:00 | 2017/3/14 8:45 | 223.10 | 2632.50 |
| 21 | 1 | 9 | 15 | 0.664 | 9 | 2017/3/14 11:20 | 2017/3/14 12:00 | 2017/3/14 13:00 | 2017/3/15 5:00 | 223.10 | 3159.00 |
| 22 | 3 | 8 | 13 | 0.664 | 7.5 | 2017/3/10 21:05 | 2017/3/10 21:45 | 2017/3/10 21:45 | 2017/3/13 11:15 | 223.10 | 2925.00 |
| 23 | 3 | 9 | 15 | 0.664 | 9.75 | 2017/3/13 10:35 | 2017/3/13 11:15 | 2017/3/13 11:15 | 2017/3/14 5:00 | 223.10 | 3422.25 |
| 24 | 2 | 6 | 8 | 0.8 | 9 | 2017/3/13 14:57 | 2017/3/13 15:45 | 2017/3/13 15:45 | 2017/3/14 7:45 | 280.00 | 3510.00 |
| 25 | 2 | 7 | 11 | 0.8 | 6 | 2017/3/14 6:57 | 2017/3/14 7:45 | 2017/3/14 7:45 | 2017/3/14 15:45 | 201.60 | 1872.00 |
| 26 | 3 | 10 | 17 | 0.936 | 8.25 | 2017/3/13 16:03 | 2017/3/13 17:00 | 2017/3/14 8:00 | 2017/3/15 8:15 | 235.87 | 2574.00 |
| 27 | 2 | 8 | 13 | 0.664 | 6 | 2017/3/14 15:05 | 2017/3/14 15:45 | 2017/3/14 15:45 | 2017/3/15 8:45 | 223.10 | 2340.00 |
| 28 | 2 | 9 | 15 | 0.664 | 11.25 | 2017/3/15 11:20 | 2017/3/15 12:00 | 2017/3/15 13:00 | 2017/3/16 7:15 | 223.10 | 3948.75 |
| 29 | 1 | 10 | 18 | 1.17 | 10 | 2017/3/15 3:49 | 2017/3/15 5:00 | 2017/3/15 5:00 | 2017/3/15 17:00 | 210.60 | 2400.00 |
| 30 | 2 | 10 | 17 | 0.936 | 6.75 | 2017/3/15 16:03 | 2017/3/15 17:00 | 2017/3/16 8:00 | 2017/3/16 15:45 | 235.87 | 2106.00 |

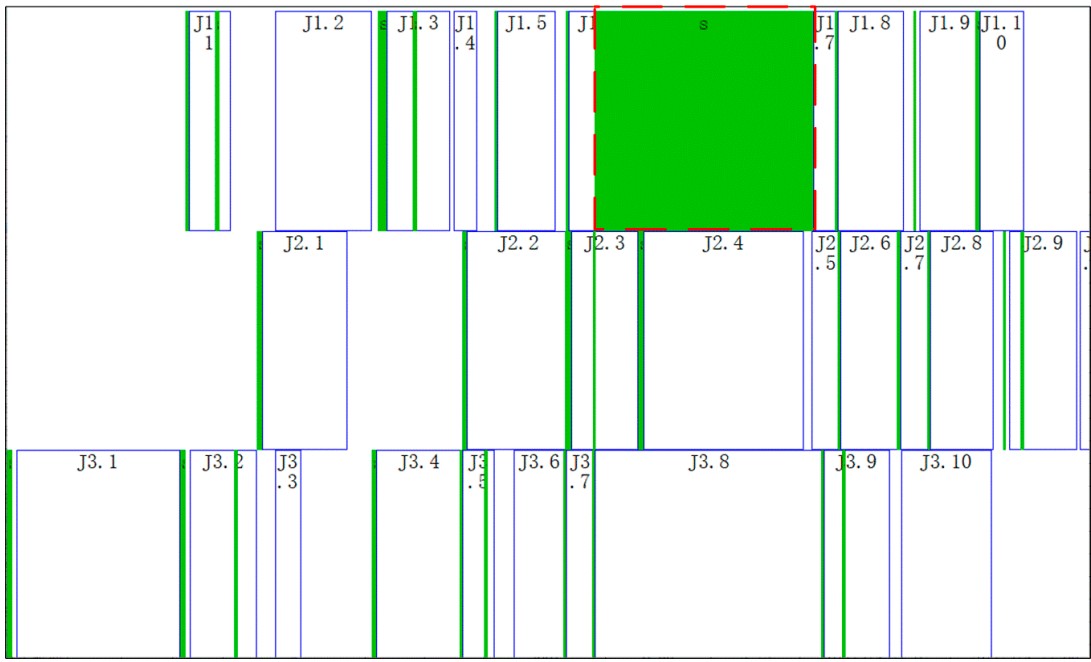

**Figure 28.** Gantt chart of the scheduled workpieces of No. 1 schedule.

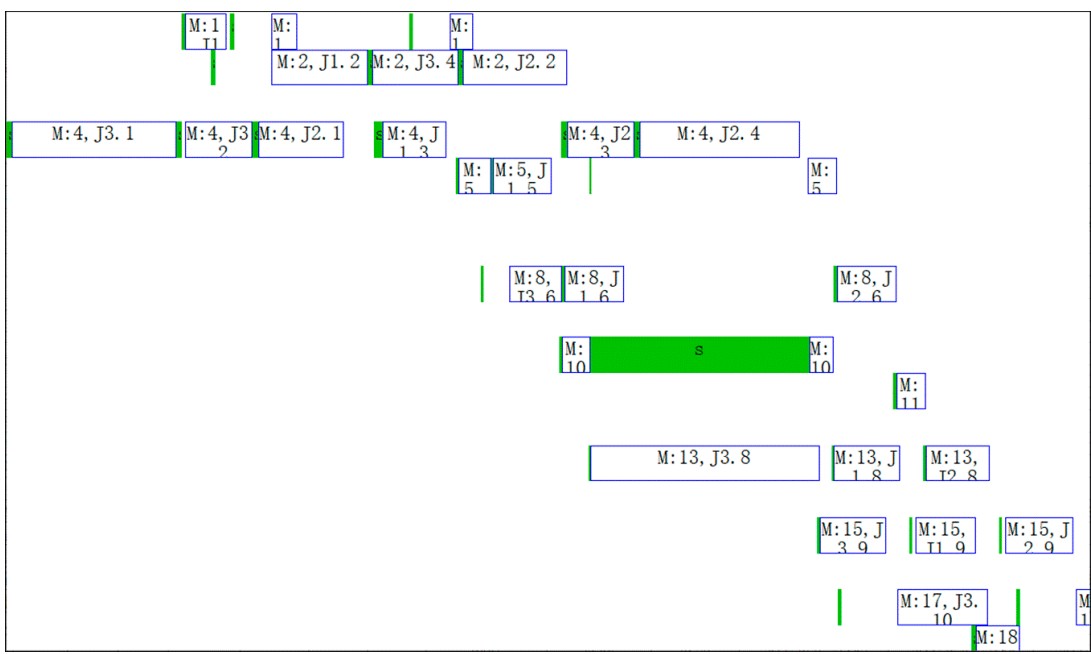

**Figure 29.** Gantt chart of the machines of No.1 schedule.

In Figures 28 and 29, the boxes with *s* identifier indicate adjusting start-end periods of processes, and the boxes with M or J identifier indicate processing start-end periods of processes. For example, the dotted box in Figure 28 indicates adjusting the start-end period of the 7th process of workpiece 1. The righthand box close to it indicates processing the start-end period of the 7th process of workpiece 1 (In row 19 of Table 8). As can be seen from row 19 of Table 8, this process is arranged on machine 10 and the adjusting start time, adjusting end time, processing start time and processing end time of this process is respectively 2017/3/10 21:45, 2017/3/13 8:23, 2017/3/13 8:23 and 2017/3/13 15:00. According to work calendar of machine 10 (M5515), we can know that calendar time between 2017/3/10 21:45 and 2017/3/13 8:23 is 58.64 h, but work time is only 0.64 h; Calendar time between 2017/3/13 8:23 and 2017/3/13 15:00 is 6.625 h, but work times are 5.625 h. 0.64 h and 5.625 h consistent with adjusting time

*st* and processing time *ct* in Table 8. This shows that the result of time reckoning according to the machines' work calendars in the algorithm is correct.

In Table 8 the sequence of processes on machine 4 (T42) is J3.1, J3.2, J2.1, J2.3, J1.3 and J2.4 can be seen to be in ascending order by No. However, it can be seen from Figure 29 that the sequence of processes on machine 4 from left to right is J3.1, J3.2, J2.1, J1.3, J2.3 and J2.4. The above two sequences are not consistent. The reason for this phenomenon is that "extrusion insertion" method is adopted in decoding operation. This method can reduce idle time of the machines as much as possible and shorten production cycle.

- No.2 schedule

The information of scheduled workpieces of No.2 schedule is shown in Table 9. The start time of No.2 schedule is 2017/3/10 8:00:00. Another condition is the same as No.1 schedule. The Pareto solution set is shown in Figure 30. Calculation time is about 40 s. The dispatcher can select one of the Pareto solutions to arrange production. The Gantt chart of the scheduled workpieces and Gantt chart of the machines of the Pareto solution (Production cycle is 16.64 days and total cost is 131,805.00 yuan) are shown in Figures 31 and 32. After the No.1 and No. 2 schedules, the Gantt chart of the machines is shown in Figure 33.

**Table 9.** The information of the scheduling workpieces of No.2 schedule.

| Workpiece No. | Workpiece Name | Type | Total Process Number | Delivery Date | Earlier Completion Rate (yuan /d) | Delayed Completion Rate (yuan/d) |
|---|---|---|---|---|---|---|
| 2 | G46-100F | L5 | 10 | 2017/4/15 | 150 | 1200 |
| 4 | L90GF | M2 | 10 | 2017/4/20 | 120 | 1500 |
| 5 | L35MC | X5 | 10 | 2017/4/13 | 70 | 600 |
| 6 | HP6100 | Y2 | 10 | 2017/4/16 | 50 | 500 |

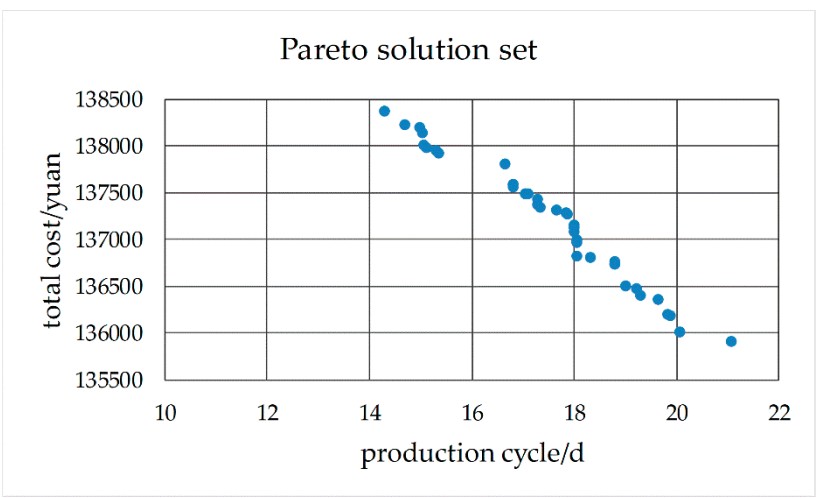

**Figure 30.** Pareto solution set of No. 2 schedule.

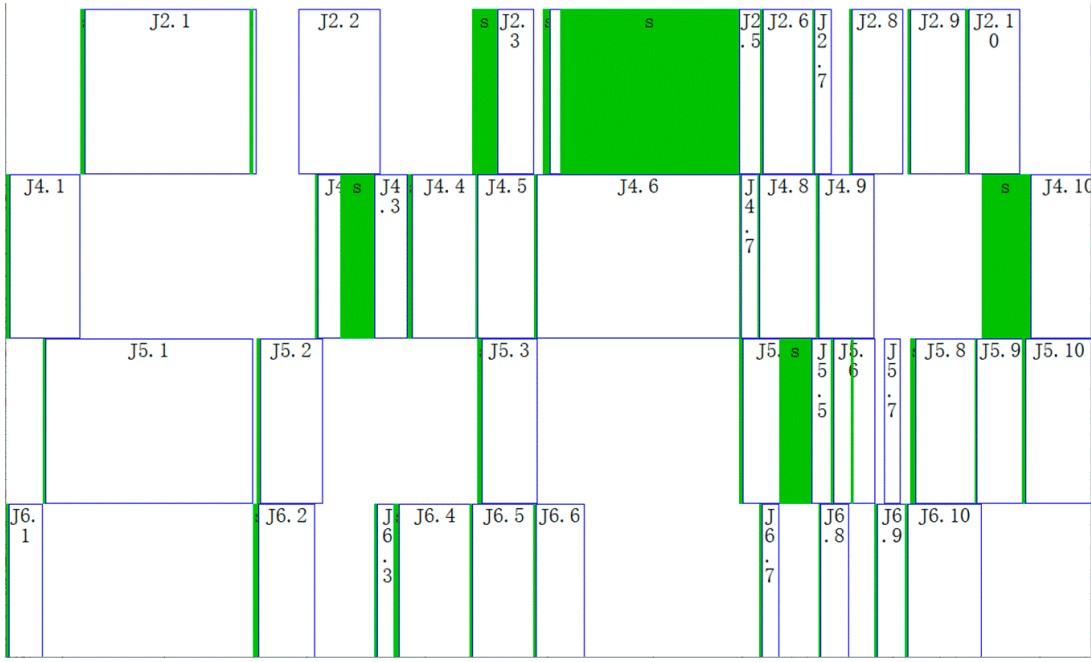

**Figure 31.** Gantt chart of the scheduled workpieces of No.2 schedule.

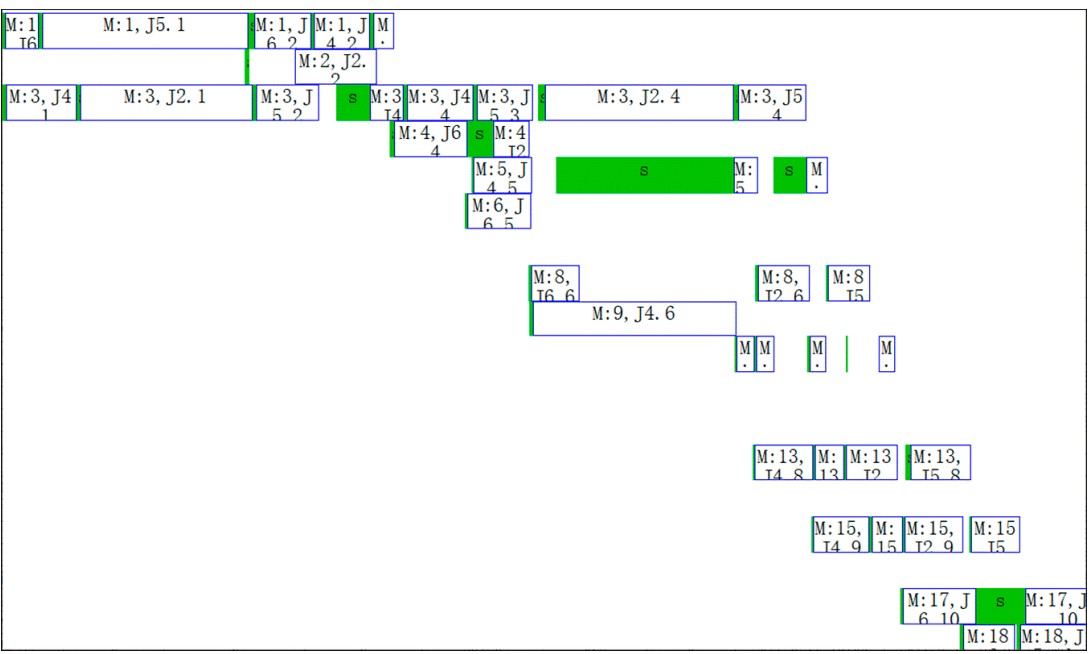

**Figure 32.** Gantt chart of the machines of No.2 schedule.

The dotted box I is a part of the Gantt chart of the No.1 schedule. The dotted box II is the Gantt chart of the No.2 schedule. It can be seen from Figure 33 that this is a typical sequential scheduling process based on the machines' work calendars.

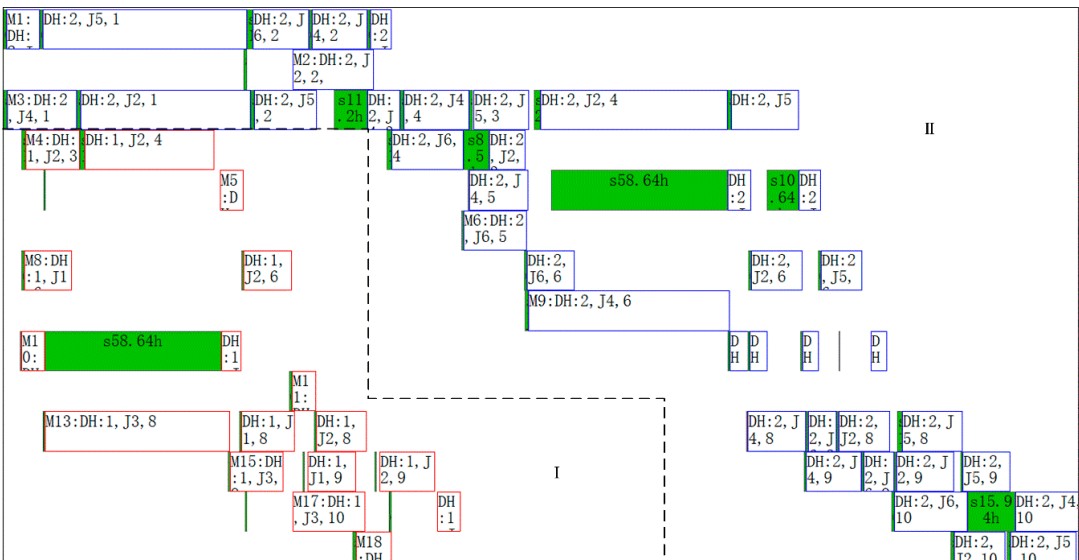

**Figure 33.** Gantt chart of the machines after No.1 and No.2 schedule.

## 7. Discussion

Because NSGA-II has a fast non-dominated sorting algorithm, elite strategy, congestion degree and congestion degree comparison operator, it retains the best of all individuals in the offspring, which improves the precision of the optimization result, the operation speed and robustness of the algorithm.

However, the case study shows that the speed of operation needs to be further increased, and the calculation time is further reduced for actual production. The algorithm considers the mixed work calendars phenomenon in actual production, so that it is consistent with reality. Meanwhile, the algorithm results are more practical, and the method can also shorten the production cycle. The scheduling scheme proposed in this paper is not the best, but it is most suitable for the actual production process.

The next step is to continue to optimize the algorithm, further reduce the computation time and algorithm complexity, and obtain a better Pareto solution set.

## 8. Conclusions

In order to solve the sequential scheduling problem for FJSP with multi-objectives under mixed work calendars, an optimization method based on NSGA-II is proposed. The research conclusions are as follows.

- The algorithm can ensure feasibility of the scheduling scheme by accurately calculating adjusting start and end time, processing start and end time of each process through time reckoning functions in the process of arranging processes.
- The algorithm can increase the utilization rate of the machines in the sequential scheduling by the following two technologies. One is taking the time status of the machines of the previous schedule as the initial status of the machines of the next schedule. The other is adopting the "extrusion insertion" method to arrange each process.
- The proposed method can help to obtain an effective Pareto solution set of the sequential scheduling problem for FJSP with multi-objectives under mixed work calendars for the dispatcher to make decisions in acceptable time.

**Author Contributions:** Q.Z. designed and performed the experiments; L.S. provided analysis software; H.S. and M.W. analyzed the data; M.W. organized the data and wrote the paper.

**Funding:** This research was funded by the Research Fund for universities of Henan Province (grant number: 19A410001), and the Research Fund for Doctoral Program of Henan Polytechnic University (grant number: B2011-088).

**Conflicts of Interest:** The authors declare no conflict of interest.

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
