# Peer review of "Sequential Scheduling Method for FJSP with Multi-Objective under Mixed Work Calendars"

_processes, doi:10.3390/pr7120888_

Round 1
Reviewer 1 Report
The study "Sequential scheduling method for FJSP with multi3 objective under mixed work calendars" presented by authors is seems a original research. However the quality of presentation is of a low level. The proposed method can help the sequential scheduling problem with multi-objective under mixed work calendars.
Below are my generic comments:
The paper is too wordy and have too many unwanted Figures. Formatting of text is not consistent. Overall the resolution of Figures are extremely poor. For example, unable to read figure 27 to 37. It seems like a project report is simply submitted as a paper without proper formatting of journal requirements.
In my opinion, the content is publishable with major revision as per aforesaid points.
Additionally, kindly consider adding a discussion section before conclusion. In discussion section, explain the the shortcomings of NSGA and NSGA-II and how your proposed sequential scheduling is identified as a better alternative.
Wish you all the best!
Author Response
Dear reviewer:
Thank you very much for your letter. After reading your comment and request, we have made extensive modification on the original manuscript. Here, we attached revised manuscript for your approval. Some of your questions were answered below.
Point 1: The paper is too wordy and have too many unwanted Figures.
Response 1: The flow charts clearly explain to the reader the operation of the functions and algorithm. Only words that explain the text will make the content too vague. So, We did not delete the figures in the first half of the paper. We deleted the No. 3 scheduling scheme in the case study because No. 1 and No. 2 were sufficient to prove the feasibility of the algorithm.
Point 2: Overall the resolution of Figures are extremely poor. For example, unable to read figure 27 to 37.
Response 2: We had submitted new figures with 500 dpi. The new Gantt charts only retain key data and show the focus I want to express.
Point 3: Additionally, kindly consider adding a discussion section before conclusion.
Response 3: We added a discussion section to the paper. This section includes some of the advantages of NSGA-II over NAGA, the shortcomings of the algorithm in the experiment and our next research direction.
Point 4: Does the introduction provide sufficient background and include all relevant references?
Response 4: We updated some of the references in recent years in order to provide more cutting-edge information.
Point 5: Is the research design appropriate?
Response 5: We are very sorry for our negligence. But We were confused about the inappropriate design of the research. We hope the teacher will give some more detailed suggestions. Thanks for your trouble.
Thank you for the kind advice.
Sincerely,
Wang Menghua
Reviewer 2 Report
The authors have studied a sequential scheduling method applied to scheduling problem. In overall, they have produced highly technical report of studies but it have some value to publish as a journal article. I propose that presentation of the article should be considered again and it should be more focused. For example, gantt-graphs that reader cannot read due to weak quality, are not very relevant for the article. So think what you want to present and focus on clarity. References of this article are moderately old, and even if literature is fundamental and produced decades ago, some latest trends could be useful for the readers point of view.
Author Response
Dear reviewer:
Thank you very much for your letter. After reading your comment and request, we had made extensive modification on the original manuscript. Here, we attached revised manuscript for your approval. Some of your questions were answered below.
Point 1: Are the results clearly presented?
Response 1: We had submitted new Gantt charts with 500 dpi. We deleted the No. 3 scheduling scheme in the case study because No. 1 and No. 2 were sufficient to prove the feasibility of the algorithm. Additionally, we added a discussion section before conclusion. This section includes some of the advantages of NSGA-II over NAGA, the shortcomings of the algorithm in the experiment and our next research direction.
Point 2: For example, gantt-graphs that reader cannot read due to weak quality, are not very relevant for the article. So think what you want to present and focus on clarity.
Response 2: We had submitted new figures with 500 dpi. Gantt charts can more intuitively show the results of sequential scheduling,so we kept them. But the new Gantt charts only retain key data and show the focus we want to express.
Point 3: References of this article are moderately old, and even if literature is fundamental and produced decades ago, some latest trends could be useful for the readers point of view.
Response 3: We updated some of the references in recent years in order to provide more cutting-edge information.
Thank you for the kind advice.
Sincerely,
Wang Menghua
Round 2
Reviewer 1 Report
The author has revised the manuscript as per suggestions. The manuscript can be now accepted.
Reviewer 2 Report
Thank you for modifications, article is adequate for the publishing.